# Perineuronal nets in HVC and plasticity in male canary song

**Gilles Cornez**[1], **Shelley Valle**[1], **Ednei Barros dos Santos**[1], **Ioana Chiver**[1], **Wendt Müller**[2], **Gregory F. Ball**[3], **Charlotte A. Cornil**[1], **Jacques Balthazart**[1] *

**1** Laboratory of Behavioral Neuroendocrinology, GIGA Neurosciences, University of Liege, Liege, Belgium, **2** Behavioural Ecology and Ecophysiology Lab, University of Antwerp, Antwerp, Belgium, **3** Department of Psychology, University of Maryland, Maryland, College Park, MD, United States of America

* jbalthazart@ulg.ac.be

**Data Availability Statement:** All relevant data are within the manuscript and its Supporting information files. Additional data was deposited on Dryad doi: 10.5061/dryad.5mkkwh76k. Prior to publication, you may find the dataset using this

## Abstract

Songbirds learn their vocalizations during developmental sensitive periods of song memorization and sensorimotor learning. Some seasonal songbirds, called open-ended learners, recapitulate transitions from sensorimotor learning and song crystallization on a seasonal basis during adulthood. In adult male canaries, sensorimotor learning occurs each year in autumn and leads to modifications of the syllable repertoire during successive breeding seasons. We previously showed that perineuronal nets (PNN) expression in song control nuclei decreases during this sensorimotor learning period. Here we explored the causal link between PNN expression in adult canaries and song modification by enzymatically degrading PNN in HVC, a key song control system nucleus. Three independent experiments identified limited effects of the PNN degradation in HVC on the song structure of male canaries. They clearly establish that presence of PNN in HVC is not required to maintain general features of crystallized song. Some suggestion was collected that PNN are implicated in the stability of song repertoires but this evidence is too preliminary to draw firm conclusions and additional investigations should consider producing PNN degradations at specified time points of the seasonal cycle. It also remains possible that once song has been crystallized at the beginning of the first breeding season, PNN no longer play a key role in determining song structure; this could be tested by treatments with chondroitinase ABC at key steps in ontogeny. It would in this context be important to develop multiple stereotaxic procedures allowing the simultaneous bilateral degradation of PNN in several song control nuclei for extended periods.

## Introduction

Songbirds must be exposed to adult tutor songs during a sensitive period in their ontogeny to be able to sing a normal species-specific song in adulthood [1, 2]. This stage of song memorization is followed by a sensorimotor learning phase during which song production is progressively refined ultimately leading to the production of the adult typical "crystallized" song [3]. Song learning and production rely on a set of interconnected brain nuclei called the song

temporary link: https://datadryad.org/stash/share/
a8yGi7AP5ZEl7FzKEjyLUUSthGXMqjwQEk_
r11VG8G4.

**Funding:** This work was supported by -a grant
from the National Institute of Neurological
Disorders and Stroke (RO1NS104008) to GFB, JB
and CAC -a grant (FSR-S-SS-17/26) from the
Special Fund for Research from the University of
Liège to CAC -CAC is FRS-FNRS Senior Research
Associate -GC was Research Fellow of the FRS-
FNRS. The funders had no role in study design,
data collection and analysis, decision to publish, or
preparation of the manuscript.

**Competing interests:** The authors have declared
that no competing interests exist.

control system that contains two main pathways. The first one connects directly HVC (used as
a proper name) to RA (nucleus robustus of the arcopallium) and is involved in song produc-
tion. The second one connects indirectly HVC to RA through, among others, Area X from the
basal ganglia. This pathway is involved in song learning and generating song variability [4–7].

Interestingly, some songbird species, such as canaries (*Serinus canaria*) or European star-
lings (*Sturnus vulgaris*), called open-ended learners [8, 9] are able to modify their song during
adulthood: they recapitulate the sensorimotor song learning stage and in some species there is
even a seasonally recurrent period of song memorization taking place on a seasonal basis [10–
12]. Canaries sing longer songs at higher amplitude that include fast repetitions of syllables
(trills) and produce specific song elements called sexy syllables at higher rate during the breed-
ing season than in the fall when they produce plastic song [13–15]. Additionally, their syllable
repertoire changes between successive breeding seasons [14, 16], but we do not know whether
new syllables are learned in adulthood during periods of song plasticity or simply reflect the
expression of different parts of the repertoire memorized during development. To our knowl-
edge, learning of new song elements in adulthood has only been formally demonstrated in
European starlings [11].

These seasonal changes are mainly driven by seasonal changes in the photoperiod and that
result in changes in blood testosterone concentrations [17, 18]. During the late summer, when
days are long, these birds are photorefractory (they no longer exhibit physiological responses
to long daylengths), their gonads are regressed and their testosterone blood concentration is
low. During the fall, when day lengths are short, they become photosensitive again (able to
respond to long daylengths with a physiological response) and later during the winter and
early spring the gonads regrow and testosterone concentrations increase to reach their highest
level in the spring during the breeding season [19–21]. We recently confirmed this suite of
endocrine and behavioral events in male canaries that were held indoors and exposed during
two annual cycles to changes in photoperiod that mimic natural changes observed at 50.6°
North, the latitude of Belgium [22].

A host of studies have demonstrated the existence of seasonal changes in song control
nuclei including changes in their volume [23], in their connectivity [24], in the number and
rate of replacement of HVC neurons [25, 26], in the size and spacing of neurons in RA and
area X [27] and in spine density of RA neurons [28], but many significant questions remain
concerning the neural bases of this seasonal song plasticity.

It was recently suggested that perineuronal nets (PNN) could play a key role in the control
of these seasonal changes. PNN are aggregation of components of the extracellular matrix
including chondroitin sulfate proteoglycans, tenascin R and hyaluronic acid that form a scaf-
fold mainly around parvalbumin-expressing interneurons [29]. They have been shown to limit
synaptic and behavioral plasticity in several mammalian species that serve as model systems
for neuroplasticity, including visual cortex plasticity [30, 31] and various forms of learning and
memory [32–36].

In zebra finches (*Taeniopygia guttata*), PNN density increases in the song control nuclei
during ontogeny to reach their highest density when sub-adult males crystallize their song [37,
38]. Females in this species never sing and correlatively have a much lower density of PNN in
all song nuclei [39, 40]. Additionally, we showed that in canaries PNN numbers also increase
in the song control nuclei during the song crystallization process that takes place during the
winter in juvenile males [41]. Furthermore, in adult male canaries, the number of PNN in
song control nuclei is lower during the fall when song is plastic than in the spring when song is
crystallized [22]. These changes in PNN numbers are correlated to and controlled, at least in
part, by changes in circulating testosterone concentrations. Accordingly treatment with exoge-
nous testosterone increases their number in HVC, RA and Area X of both males and females

[42]. Based on these data, we hypothesized that changes of PNN density represent a significant mechanism regulating seasonal vocal plasticity in songbird species that exhibit the pattern of adult open-ended learners.

Degradation of PNN via the application of the enzyme chondroitinase ABC (ChABC) has been shown in mammals to restore brain and behavioral plasticity in a variety of experimental conditions [31, 33–36]. Interestingly, two preliminary studies, published in abstract form only, suggested that dissolution by ChABC of PNN in the male zebra finch HVC tends to increase song variability [43, 44].

To test the hypothesis of a causal role of PNN in the control of song plasticity, we degraded PNN in the HVC of male canaries during three independent experiments and quantified the resulting changes in vocal behavior. To maximize the probability of detecting effects of PNN degradation, these effects were assessed in different endocrine conditions namely a) in the fall when birds were beginning a new singing season and their singing activity was stimulated with exogenous testosterone (Experiment 1), b) in the absence of exogenous testosterone when birds were spontaneously singing at the end of the breeding season (Experiment 2) and c) at the height of the breeding season (Experiment 3). In addition, during two experiments, experimental males were exposed to the playback of unknown songs to increase the possibility of changes in song structure.

## Methods

### Subjects

Three separate experiments were performed with male canaries of the Fife Fancy breed that originated from the outbred laboratory-based population kept at the Behavioral Ecology and Ecophysiology lab of the University of Antwerp, Belgium. Prior to the transfer, all birds were molecularly sexed [45] and only birds that were not closely related (i.e., that originated from different broods) were selected for the experiments. Upon arrival, all birds were maintained in an indoor aviary or in collective cages containing 4 to 7 birds under an artificial photoperiod that was adjusted monthly to the outside photoperiod in Liège (50.6 N). We ensured in this way that males would display the spontaneous seasonal variations in singing activity that are typical for this canary breed, as quantitatively described previously [22]. For song recording sessions, birds were moved to individual cages each located inside a sound-attenuated chamber and exposed to the same photoperiod they experienced in the aviary or collective cages. All birds received food and water *ad libitum* throughout the experiments. All experimental procedures complied with Belgian laws concerning the Protection and Welfare of Animals and the Protection of Experimental Animals. Experimental protocols were approved by the Ethics Committee for the Use of Animals at the University of Liège (Protocol 1739).

### Experimental design

During each of these three experiments, the PNN present in HVC were temporarily degraded by application of the enzyme chondroitinase ABC (or saline as a control solution) to a region on the dorsal edge of the nucleus by a procedure initially developed by the laboratory of Teresa Nick at the University of Minnesota [37]. Birds were anesthetized with isoflurane (3% induction, 1.5% maintenance) and fixed in a stereotaxic apparatus (Kopf Instruments; Tujunga, CA, USA). An incision was made in the head skin and the skull was exposed. A 2 by 2 millimeters square window was opened in the skull of each hemisphere dorsal to HVC (1.5 to 3.5 mm in the medio-lateral and 0.0 to -2.0 mm in the antero-posterior direction relative to the reference point). A 2 by 2 mm piece of gelfoam that had pre-absorbed 5 μl of chondroitinase ABC (ChABC, from *Proteus vulgaris*, Sigma C3667, diluted at 50 U/ml) or of saline solution (control

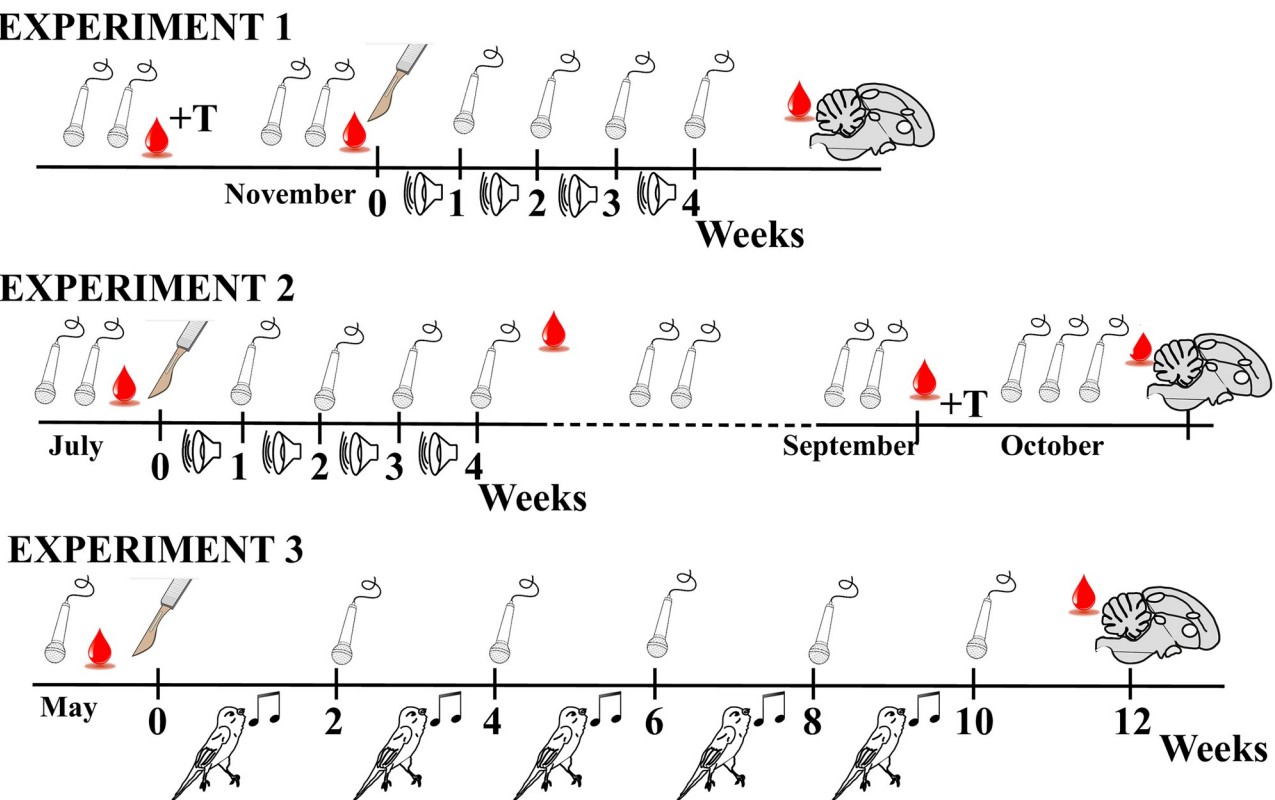

**Fig 1. Schematic representation and comparison of the experimental designs implemented during experiments 1 to 3.** The different symbols present in the figure represent recording sessions (microphone), tutoring with song playback (loudspeaker), tutoring with live male tutors (bird singing), surgeries and application of ChABC or saline over HVC (scalpel), subcutaneous implantation of a Silastic™ capsule filled with testosterone (+T), collection of blood sample (red drop of blood) and collection of the brain at the end of the experiment (sagittal section through brain).

birds) was applied on the dorsal edge of each hemisphere in the skull windows that were then closed with dental cement and the skin was sutured with a 5–0 coated Vicryl™ thread. Birds were allowed to recover in an individual cage under a warm lamp where they recovered (i.e., moving and perching normally) quickly.

The birds singing activity was recorded on selected days during the next few weeks or months depending on the experiment and its specific goal. Songs were recorded for 2 hours immediately after lights on when their singing activity is most intense. We first describe the specific aspects of each experiment (see Fig 1) before presenting the methodological aspects that are common to all of them.

### Experiment 1

For the first experiment, 16 adult male canaries that were approximately 2.5 years old were obtained from the University of Antwerp in late October. On November 1st, 16 of these birds were transferred to sound-attenuated chambers where they remained until the end of the experiment to record their singing behavior. A subcutaneous testosterone-filled Silastic™ implant was inserted under their skin on November 4th to induce song crystallization as well as enhance singing activity (see next sections for more detail).

When all 16 recorded birds sang crystallized song, surgeries inducing the degradation of PNN in HVC were carried out from November 22nd to 24th: 9 birds received a bilateral treatment with ChABC while 7 birds received a bilateral treatment with a control solution. One

bird from the ChABC group died a few days after surgery and various technical problems caused loss of data thus reducing sample size to 7 Ctrl and 8 ChABC males for song analyses and to 6 Ctrl and 6 ChABC males for the analysis of morphological and physiological data (brain, periphery and plasma T concentrations). Four additional males obtained from a local dealer were also treated unilaterally with ChABC in the same way and their brain was collected one (n = 2) or three (n = 2) days later to analyze the PNN depletion by immunohistochemistry. The treated hemisphere was counterbalanced across the three subgroups.

The 16 bilaterally treated birds were returned to the sound-attenuated chambers to continue recording their singing behavior (2 hours of recording on weeks 1, 2, 3 and 4 after surgery). They were additionally exposed to tutor songs playback to test whether the PNN degradation facilitates incorporation of new syllables into the repertoire. The playback tapes contained songs of adult male canaries of the Border breed that sing mainly different syllable types compared to the Fancy Fife breed, thus facilitating the identification of novel syllables. The tapes were provided by the Department of Psychology, University of Maryland in College Park. The playback exposure took place inside the recording boxes and lasted 2 hours per day during the afternoon when no recordings were ongoing. After 4 weeks of recordings and tutor exposure, all brains were collected and fixed. Blood samples were taken from the wing vein of each bird in the morning on the days of implant insertion, surgery and brain collection.

## Experiment 2

The second experiment was carried out with 15 one year-old male canaries that had been kept under natural photoperiodic conditions prior to arrival at the laboratory in Liege in mid March. This second experiment was planned to extend results of the previous experiment by following the spontaneous changes in song structure that take place across the annual cycle instead of using testosterone implants to artificially enhance singing activity and song crystallization. The use of T implants might indeed explain the limited effects of PNN degradation that had been observed. In our laboratory, canaries sing crystallized song during the spring and early summer, after which the song becomes more plastic. Initially surgeries were performed on July 2 to 4 after the birds' vocalizations had been recorded on 2 separate days for 2 hours a day (pretests 1 and 2, PT1 and PT2). 7 birds were bilaterally treated with ChABC and 8 birds were similarly treated with the control solution. These birds had been matched for their singing performance before surgery but unfortunately 6 Ctrl and 2 ChABC birds died soon after surgery for unexplained reasons (possibly due to the mite infection that was present in the animal facility at that time). Additional birds (5 Ctrl and 3 ChABC) went through the same surgery during the next few days (final sample size 7 Ctrl and 8 ChABC) but unfortunately only two of these replacement birds (ChABC treated) were singing regularly. The usable sample size was thus reduced for the recordings performed during the next 4 weeks to 7 ChABC but only 2 Ctrl birds. The Ctrl birds can thus only serve as a qualitative reference point. Birds were recorded and exposed to tutors in the same way as in the first experiment until early August. Their song was then recorded for 2 hours once a week on weeks 1, 2, 3 and 4 after the surgery (T1 to T4).

All birds were then left undisturbed in group cages until the early autumn when canaries usually start singing plastic songs. By mid-September, birds were transferred back to the recording chambers and their vocalizations were recorded until the end of October. Because very little singing activity was detected during 8 recording sessions spread between September 11 and October 3, a testosterone implant was inserted subcutaneously in each subject on October 3. This progressively enhanced singing activity that was recorded during 13 two-hour sessions (on October 4, 5, 6, 7, 8, 10, 11, 12, 14, 15, 16, 20, 29) until October 29 when it reached a

nearly stable level. Song recordings were analyzed in detail for two of the dates before testosterone implantation (September 28 and October 2, respectively Baseline (BL) 1 and 2) and at four time points after implantation (October 8, 14, 20 and 29, respectively T1 to T4). The brain of all these subjects was collected at the end of October, almost 4 months after surgeries to explore their rate of PNN reconstruction after experimental degradation.

Blood samples were collected 2 weeks before surgeries (Pretest; PT), 5 weeks after surgeries after the 4 weeks of song recording (T5), in early October one week before testosterone implants insertion (Pre T) and at the end of the experiment on the day of brain collection (End).

## Experiment 3

This experiment was started with a total of 30 males that were over 2 years old that had been so far exposed to the natural photoperiod. They were housed as a group in an indoor aviary until early May and at that time they were placed individually in the recording chambers and their singing behavior was recorded for 2 hours on 5–6 different days.

Since more birds were included in this experiment than the number of available sound-attenuated boxes (n = 16), the experiment was run in two cohorts separated by two weeks. Surgeries inducing PNN degradation by application of ChABC and the control surgeries involving application of saline were performed two weeks apart in these two cohorts in May. Each cohort contained an equal number of ChABC and control birds. Afterwards, birds did not reside permanently in the recording boxes but they were placed there only for 3–4 days every week and their singing activity was recorded for 2 hours on each of these days. Between these recording sessions, birds were kept in individual cages located in the same room where they were exposed to live male canary tutors. These tutors were 5 males of the same breed raised under the same photoperiod that were all housed in one cage placed in front of the cages holding the experimental subjects. These procedures including alternation of tutoring and song recording were continued for 10 weeks past surgeries.

Unfortunately, a substantial number of subjects died during or soon after the surgical procedures and additionally a number of subjects never sang normal songs thereafter suggesting that their HVC had been inadvertently lesioned. Final sample size was therefore reduced to 7 Ctrl and 8 ChABC treated males. Blood samples were collected before surgeries and 12 weeks later just before brains were collected for histological analyses in August. Songs were analyzed in detail in 2 hours of recording for each bird collected before (T0) and 2, 4, 6, 8 and 10 weeks after (T2 to T10) surgeries. All other procedures described in the following sections were common to all experiments.

## Testosterone implants

During experiments 1 and 2, a 10 mm long Silastic™ implant (Dow Corning reference no. 508–004; inner diameter 0.76mm, outer diameter 1.65mm) filled with crystalline testosterone (Fluka Analytical, Sigma-Aldrich) was inserted subcutaneously in the back of the birds. Before insertion, implants were incubated in 0.9% NaCl at 37°C overnight to initiate steroid diffusion. A small hole was made in the skin of the back of the bird, the implant was inserted and the hole was closed with surgical glue or sutured with a 5–0 coated Vicryl™ thread.

## Testosterone assays

Blood samples (50–150 μl) were collected from the wing vein into Na-heparinized micropipettes and directly centrifuged at 9000 g for 9 minutes. Plasma was separated and stored at

-80˚ C until assayed. Blood collection was always performed within 3 minutes after catching the birds in their cage during the morning, within 1.5 h after lights on.

Testosterone was assayed with an Enzyme Immunoassay (EIA) kit from Cayman Chemicals (Ann Harbor, Mi USA, Item No. 582701) following manufacturer's instructions as previously described [22, 41]. Briefly, 10 µl of plasma from each sample was diluted in 150 µl of ultra-pure water. Recovery samples were spiked with 20,000 CPM of tritiated-testosterone (Perkin-Elmer). All samples were extracted twice with 2 ml of dichloromethane. The organic phase was eluted into clean tubes, dried with nitrogen gas and stored at -20˚C until further use. All samples from each experiment were extracted at the same time. Mean average recovery of extractions was to 81.25, 81.44 and 69.2% for Experiments 1, 2 and 3 respectively.

Extracted samples were re-suspended in 400 µl Enzyme Immunoassay (EIA) buffer and 50 µl of this buffer were assayed for testosterone in triplicate. The mean intra-assay variations for the 3 experiments were 2.6, 3.8 and 4.1% for Experiments 1, 2 and 3 respectively. The lower detection limit of the assay was around 0.1–0.2 pg/well corresponding to 80–160 pg/ml.

## Song recording

Song recordings were always obtained during the first 2 h immediately following lights-on inside 16 custom-built sound-attenuated boxes where birds were singly housed. Sound was acquired from all 16 channels simultaneously via custom-made microphones (microphone from Projects Unlimited/Audio Products Division, amplifier from Maxim Integrated) and an Allen & Heath ICE-16 multichannel recorder connected to a computer. The 16-bit sound files were acquired and saved as 1 min.wav files by Raven v1.4 software (Bioacoustics Research Program 2011; Raven Pro: Interactive Sound Analysis Software, Version 1.4, Ithaca, NY: The Cornell Lab of Ornithology) at a sampling frequency of 44,100 Hz that translates into frequencies in the range 0 to 22,050 Hz.

## Song analysis

The daily 2 hour sound recordings were first reassembled for each channel corresponding to each experimental bird. These song files were then analyzed by a custom-made Matlab software designed to detect and analyze canary songs and syllables into recordings that was developed and provided by Ed Smith in the laboratory of Prof. Robert Dooling, University of Maryland. The suitability of this program for analyzing canary songs was previously validated [46]. The program defined a vocalization as a song if it was at least 1 s long, was preceded and followed by at least 0.4 s of silence and was at least 30 dB above background noise.

The software determined the song rate (number of songs per hour) and the percentage of time spent singing. It also computed for each separate song a large number of measures including song duration, song Root Mean Square (RMS) power (a measure of song loudness), song entropy (a measure of disorder within power distribution across frequencies), song bandwidth (range of frequencies of the songs that are produced), power distribution across the 1st, 2nd and 3rd quartile frequencies, number of syllables per song, average syllable duration per song and average syllable RMS power per song. For each parameter, the values corresponding to all songs from a single channel (bird) on a single day were averaged and a coefficient of variation was calculated. Some birds did not sing at all on specific days and had therefore to be excluded from of the analyses of song characteristics, except of course for song rate and time spent singing that were assigned a zero value. We previously showed that values produced by the Matlab software are closely correlated to manual measures of the same songs obtained by quantification of the song spectrograms prepared by the RavenPro software [46].

One prominent feature of crystallized songs is the presence of rapidly repeated highly stereotyped syllables also called trills. Another part of the Matlab software was specifically developed to detect and analyze these trills. Trills within each song were defined by the program based on multiple characteristics including the minimum distance (time) between iterations (3 msec) within a trill and distance with other syllables, the minimum number of identical segments (4) and various measures of changes in power and entropy. Based on these detection criteria, the software quantified the trill rate (number of trills per song), percentage of time spent trilling within each song, the average trill duration (in msec), the duration of each segment (in msec), the interval between trills and between each segment within a trill (in msec), the trill center frequency (in Herz), the trill bandwidth (in Herz) and finally the trill entropy. This analysis of trills was carried out for recordings of experiments 2 and 3.

In addition, manual analyses were performed during experiment 1 and 2 to assess changes in repertoire with the use of the Raven Pro v1.4 software. Spectrogram views were constructed with a direct Fourier transform (DFT) size of 256 samples (172 Hz per sample) and a temporal frame overlap of 50% with a hop size of 128 samples. These parameters were automatically determined by the software to provide an optimized frequency/time resolution.

We only analyzed crystallized songs because the variability between syllable renditions within plastic songs could bias the syllable identification. The catalogue of different syllables sang by each bird at different time points was constructed. The different time points were compared for each bird and identical syllables appearing in different songs were given a similar code name. The catalogue of syllables composing the tutor playback songs was constructed in the same way and compared to the catalogue of syllables of each experimental bird at weeks 2 and 4 after the surgery. These data were used to compute the repertoire size (total number of syllables in the catalogue) for each time point. For the 2 post-surgery time points, the number of dropped and added syllables compared to the pre-surgery catalogue was computed for each bird; the number of syllables similar to a tutor syllable was also calculated. The syllables that were similar to a tutor syllable were separately counted.

This type of analysis was not performed in experiment 3 because the tutors in this case were birds of the same breed raised in the same laboratory (Antwerp University). It was thus impossible to determine whether added syllables were learned in adulthood. Exposure to other adults as tutor still made sense, however, based on a previous study that identified more song plasticity after PNN degradation in male zebra finches exposed to social contacts as compared to social isolation [43].

## Physiological status and brain analyses

**Tissue collection.** At the end of the experiments, subjects were weighed, the width and length of their cloacal protuberance was measured, a blood sample was taken from the wing vein and then birds were anaesthetized with Nembutal™ (0.04ml at 0.6 mg/ml of pentobarbital molecule). Once reflexes had stopped, birds were intracardially perfused with phosphate-buffered saline (PBS), immediately followed by 4% paraformaldehyde PBS (PFA) to fix the brain. The brain was extracted from the skull and post-fixed during 24 hours in 15 ml PFA. The syrinx was also extracted and weighed. On the following day, brains were transferred to a 30% sucrose solution. Once brains had sunk to the bottom of the vial, they were frozen on dry ice and stored at -80˚ C until used. The two brain hemispheres were separated and cut sagittally on a Leica CM 3050S cryostat into 30 μm thick sections that were distributed in 4 series of 3 wells (1st experiment) or 4 series of 2 wells (2nd and 3rd experiment). Sections were stored in anti-freeze solution at -20˚C until used.

**Immunostaining.** One well from each hemisphere of all birds was double-labeled for parvalbumin (PV) and chondroitin sulfate, one of the main components of the perineuronal nets, following a previously described protocol [22, 37, 41]. Briefly, sections were blocked in 5% Normal Goat Serum (NGS) diluted in Tris-buffered Saline (TBS) with 0.1% Triton-X-100 (TBST) for 30 minutes. They were incubated overnight at 4°C in a mixture of 2 primary antibodies diluted in TBST: a mouse monoclonal anti-chondroitin sulfate antibody (clone CS-56, 1:500 for Experiment 1 and 3 or 1:1000 for Experiment 2, C8035, Sigma Aldrich) specific for the glycosaminoglycan portion of the chondroitin sulfate proteoglycans that are the main components of the PNN and a polyclonal rabbit anti-parvalbumin antibody (1:1000; ab11427, Abcam). Sections were then incubated at room temperature in a mixture of secondary antibodies diluted in TBST. A goat anti-mouse IgG coupled with Alexa 488 (green, 1:100, Invitrogen) was used to visualize PNN staining and a goat anti-rabbit IgG coupled with Alexa 546 (red, 1:200, Invitrogen) was used to visualize PV cells. Finally, sections were mounted on slides using TBS with gelatin and coverslipped with Vectashield containing DAPI (H-1500, Vector laboratories) to confirm that PNN that were not surrounding PV-positive cells were localized around a cell nucleus.

**PNN & PV quantification.** The numbers of PV positive cells and of cells surrounded by PNN were counted in the 3 song control nuclei HVC, RA and Area X. The boundaries of the ROIs were determined based on the bright PV and/or PNN staining. Two photomicrographs were acquired in each hemisphere in 2 sections equally spaced for each ROI. These photomicrographs were obtained with a Leica fluorescence microscope with a 40X objective and fixed settings. Each photomicrograph only contained the ROIs so that quantifying the entire image always sampled a similar area. The numbers of PV, PNN and PV+PNN positive cells were consequently counted in the entire photomicrographs with Image J software (NIH, https://imagej.nih/ij) as previously described [22, 41].

For each ROI, the mean between left and right data for each section was calculated and this result was then averaged between sections to obtain the number of stained structures per counted surface in a given ROI. These numbers were converted in densities/mm$^2$ and also used to compute the % PV neurons surrounded by PNN (%PVwithPNN).

All brain analyses were made by an observer who was blind to treatment group.

## Statistics

Plasma testosterone concentrations and results of automated song analysis were analyzed by two-way repeated measures ANOVAs with time of data collection as a repeated factor and the two treatments (ChABC and ctrl) as a fixed factor. If the time effect or the interaction was significant, the Sidak's multiple comparisons test was used to explore possible differences between time points or between groups at each time point. The brain data and song parameters were analyzed using unpaired t-tests. Analyses were carried out with the GraphPad Prism Software Version 8.2.1. All results are presented by their mean ± SEM in figures and text. In addition individual values are plotted in bar graphs. Differences were considered significant for $p \leq 0.05$.

## Results

### Experiment 1

**Physiology and morphology.** Plasma testosterone concentrations were low in both groups at the beginning of the experiment (Ctrl group: 0.46±0.07, ChABC group: 0.33±0.10 ng/ml) and were markedly increased by the T implant (before surgery- Ctrl group: 5.52±1.54, ChABC group: 4.83±1.32; at brain collection: Ctrl group: 6.74±1.87, ChABC group: 3.94±1.48;

S1A Fig). A two-way repeated-measures ANOVA confirmed that testosterone implants were very effective in increasing plasma testosterone concentrations over time ($F_{2,20}$ = 16.52, $p < 0.0001$) but there was no treatment effect ($F_{1,10}$ = 0.73, $p = 0.4136$) and no interaction between treatments and time ($F_{2,20}$ = 1.03, $p = 0.3755$). The post-hoc analysis identified significant differences between the pre-implant and the two post-implant periods.

When brains were collected, no difference was found in measures of sexual development including the mean mass of the two testes that were fully regressed (Ctrl group: 5.01±2.59, ChABC group: 1.70±0.67 mg, $t_{10}$ = 1.23, $p = 0.2456$, S1B Fig), the androgen-dependent cloacal protuberance area (Ctrl group: 16.67±2.13, ChABC group: 14.29±1.30 mm$^2$, $t_{10}$ = 0.95, $p = 0.3636$, S1C Fig) and syrinx mass (Ctrl group: 25.53±1.80, ChABC group: 24.65±2.30 mg; $t_{10}$ = 0.43, $p = 0.6729$, S1D Fig).

**Effect of PNN degradation on singing behavior.** The effects of testosterone on singing behavior were largely acquired when the surgeries were performed as illustrated in Fig 2 by the song features quantified during the two per-experimental recordings even if singing rate still slightly increased between PT1 and PT2. Some song features, however, continued to change after the surgeries in the same way in both groups suggesting that T was still modifying singing behavior.

The song rate and percentage of time spent singing did not differ between groups (song rate: $F_{1,13}$ = 1.22, $p = 0.2902$; % time singing: $F_{1,13}$ = 1.04, $p = 0.3268$) but changed significantly with time (song rate: $F_{5,65}$ = 8.62, $p < 0.0001$; % time singing: $F_{5,65}$ = 7.81, $p < 0.0001$) while there was no interaction between these factors (song rate: $F_{5,65}$ = 0.67, $p = 0.6482$; % time singing: $F_{5,65}$ = 0.21, $p = 0.9573$). Post-hoc analyses indicated that the T1 time point was significantly different from all other time points (Fig 2A and 2B for detail) suggesting that the surgery decreased singing activity during the following week in all birds.

As most birds did not sing or sang very rarely during the first week post surgery (T1), song quality could not be assessed at this time point that was consequently excluded from all subsequent analyses. Starting on the second week after surgery when all birds had essentially recovered their pre-experimental singing rates, song structure was apparently similar in Ctrl and ChABC-treated males and could not clearly be distinguished from songs produced before the surgeries. Syllables that were produced during the pretest were still accurately produced after the surgeries. Fig 3 presents sonograms of songs collected in these two groups illustrating this lack of obvious change.

This lack of major changes was confirmed by quantitative analyses of song structure. The song duration did not differ between treatments ($F_{1,12}$ = 0.22, $p = 0.6471$) nor across time ($F_{4,48}$ = 2.17, $p = 0.0870$) and no interaction between these factors was detected ($F_{4,48}$ = 0.72, $p = 0.5808$, Fig 2C). Interestingly, there was a main group effect ($F_{1,12}$ = 8.70, $p = 0.0121$) on the song RMS power, associated with a significant effect of time ($F_{4,48}$ = 5.09, $p = 0.0017$) and an interaction between these factors ($F_{4,48}$ = 2.57, $p = 0.0494$). Post-hoc analyses indicated a significant difference between groups at all time points following the surgery with ChABC-treated males producing less powerful songs (Fig 2D). Other song features including song entropy, song bandwidth, and power distribution across the 1[st], 2[nd] and 3[rd] quartile frequencies were not affected by group differences. No interaction between groups and time was also found, but there was a main effect of time on song entropy and on power distribution across the 1[st], 2[nd] and 3[rd] quartile frequencies. Post-hoc analyses only identified increases with time of the power distribution across the 2[nd] quartile and 3rd quartile frequencies (see Fig 2E–2I for detail).

Analyses of the number of syllables per song and of their duration identified no treatment effect and no treatment by time interaction. However, there was an effect of time on syllable

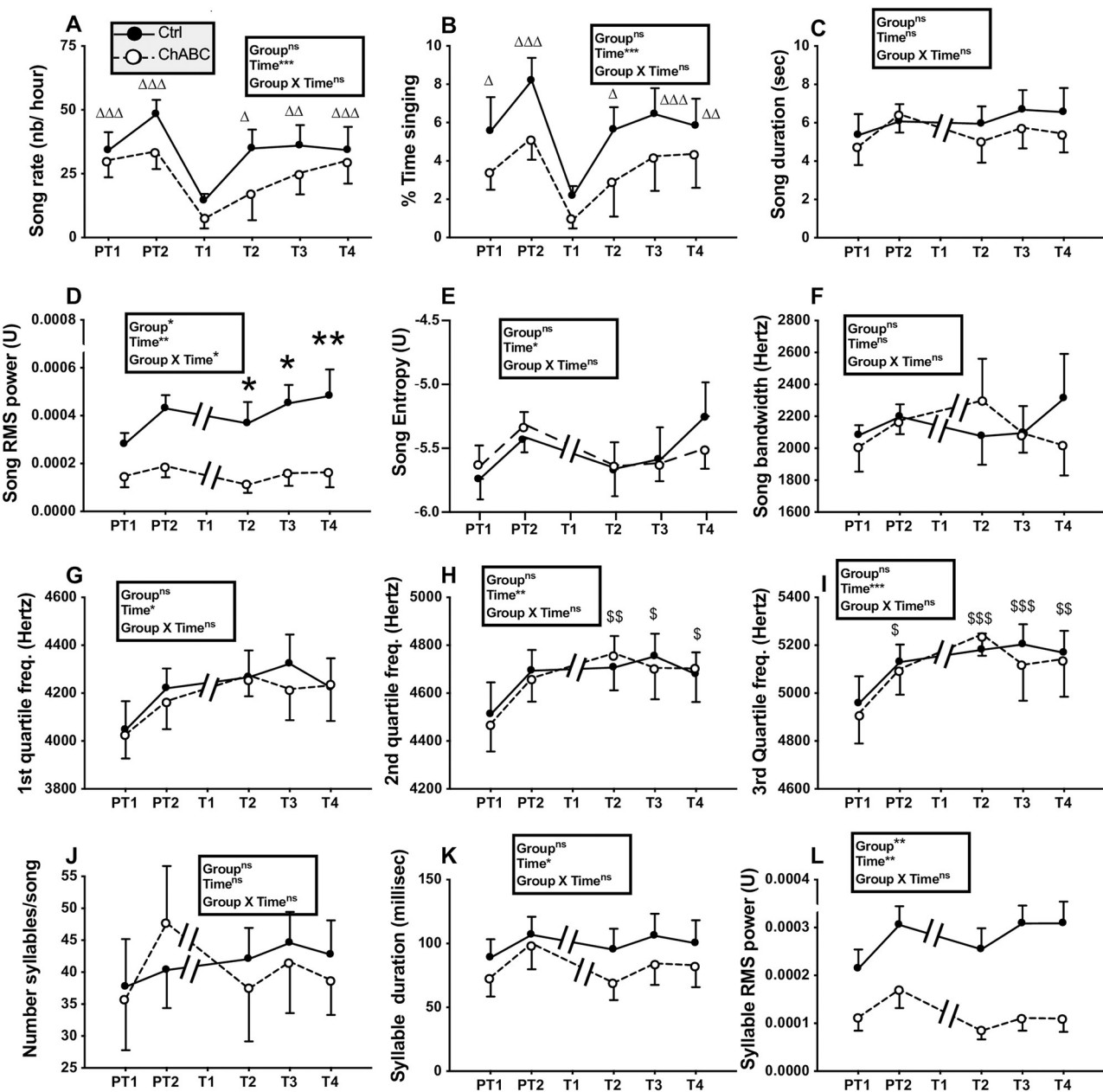

**Fig 2. Song rate and song features before (pretests PT1, PT2) and the next four weeks after (tests T1 to T4) bilateral treatment of HVC with ChABC (N = 7) or the saline control (Ctrl; N = 8).** The T1 time point was not analyzed for measurements of song quality because most birds did not sing during that week (see panels A and B). Data were analyzed by two-way ANOVA with treatments as independent factor and time as a repeated factor and results are summarized in the insert for each panel. Results of post-hoc analyses of significant interactions are indicated by asterisks (*p<0.05, ** p<0.01 compared to the Ctrl group). Results of post-hoc analyses of significant main effects of time are indicated by triangle(s) for comparison with T1 and by dollar signs for comparison with PT1 (Δ or $ p<0.05, ΔΔ or $$ p<0.01, ΔΔΔ or $$$ p<0.001).

duration although no significant difference between specific time points was detected by the post-hoc analysis (Fig 2J and 2K). There was a main effect of treatment ($F_{1,12}$ = 14.81, $p$ = 0.0023) and time ($F_{4,48}$ = 5.09, $p$ = 0.0017) on the syllable RMS power but a lack of interaction ($F_{4,48}$ = 2.31, $p$ = 0.0715), so that it cannot be ascertained that this difference between groups was not present before the surgery.

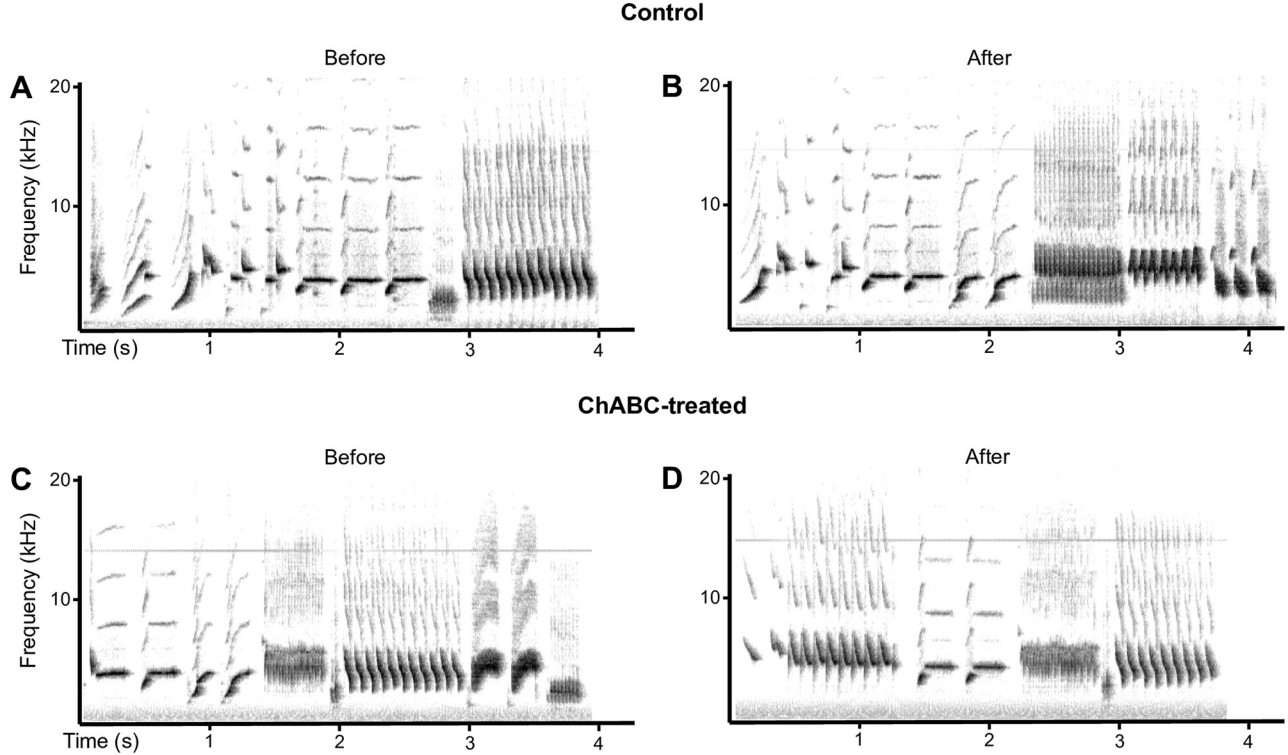

**Fig 3. Examples of sonograms of songs collected before and after surgeries in the Ctrl and ChABC-treated males.**

**Effect of PNN degradation on syllables repertoire.** It was previously shown that 200 seconds are sufficient to obtain the full repertoire composition of canary songs [47]. Since song repertoire could vary between canary breeds, we first evaluated this conclusion for the Fife Fancy breed used here by analyzing the cumulative number of different syllables detected in the song of three randomly selected males. This analysis confirmed that few or no new syllables can be detected after a total of 200 s of songs (accumulated over 20 to 35 separate songs) have been analyzed (Fig 4A). A minimum of 200 seconds of songs per bird were thus analyzed to identify the syllable repertoire of individual birds at three time points: 1 week before surgery (PT) and 2 (T2) or 4 weeks (T4) after the surgery.

Repertoire size was similar in the two groups before ($t_{12} = 0.74$, $p = 0.4731$) and 2 weeks after the surgery ($t_9 = 0.12$, $p = 0.9044$) but was significantly lower in the ChABC group compared to the control group 4 weeks after the surgery ($t_{12} = 2.22$, $p = 0.0462$, see Fig 4B). This effect would however not resist a Bonferroni correction for multiple testing. Comparison of the syllable repertoire before and after the surgery suggested that ChABC birds had dropped a higher number of syllables at T4 (but not at T2) than the control birds although this change in repertoire was quite variable between subjects and therefore no significant difference between groups was present (T2: $t_9 = 0.03$, $p = 0.9785$; T4: $t_{12} = 1.73$, $p = 0.1091$; Fig 4C). The numbers of syllables shared with the repertoire of the tutors used for playbacks was very small and did not differ between groups at the two time points after surgery (T2: $t_9 = 0.99$, $p = 0.3478$; T4: $t_{12} = 0.77$, $p = 0.4582$; Fig 4D). This was also true for the syllables added, i.e. shared with tutors after but not before surgery (T2: $t_9 = 0.23$, $p = 0.8243$; T4: $t_{12} = 0.61$, $p = 0.5557$; Fig 4E).

Finally, during T4, there was a non-significant trend of the similarity score (total number of syllables common to the pre and post repertoire divided by the total number of different

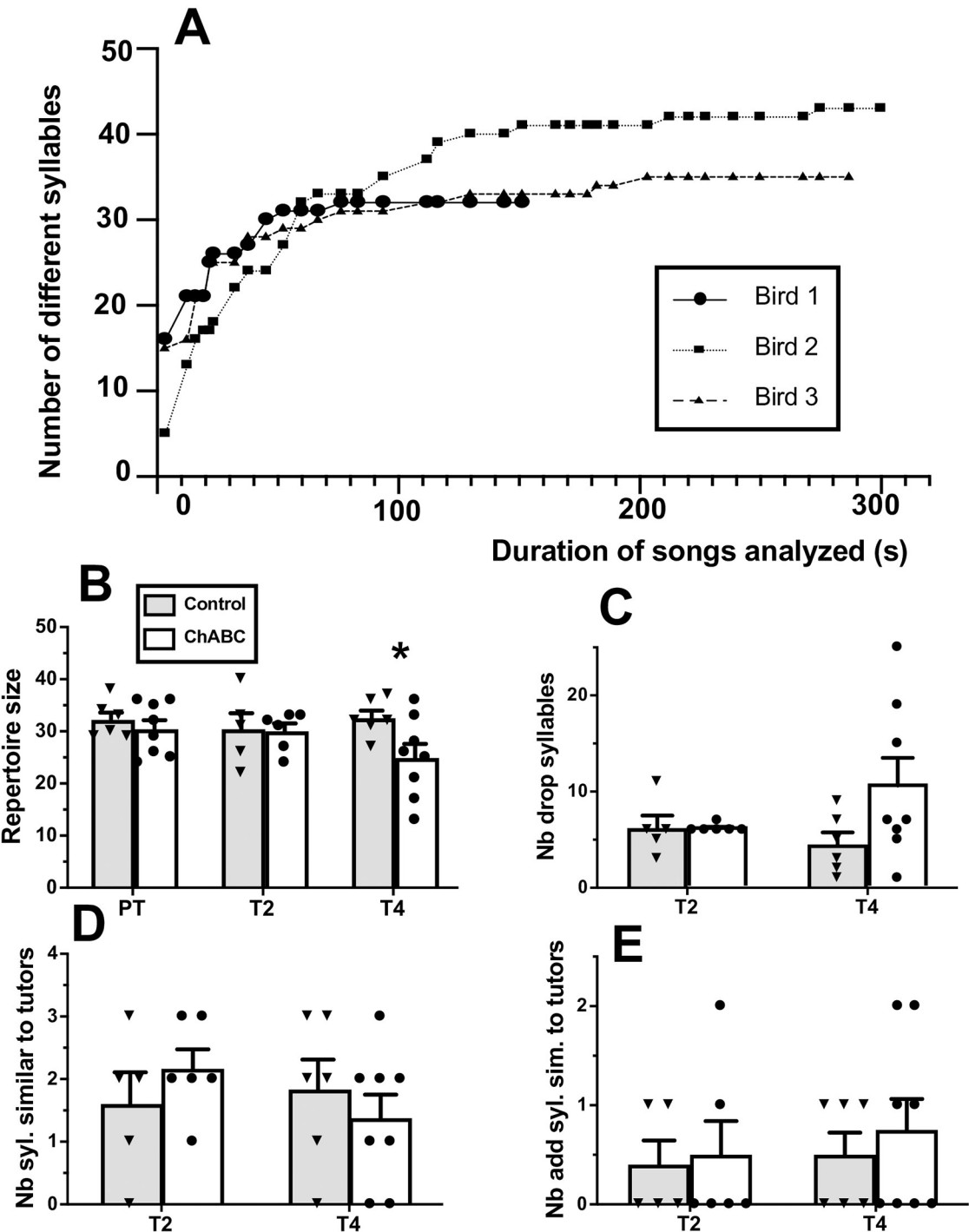

**Fig 4. Effects of ChABC on repertoire size. A.** Cumulative number of new syllables identified in the songs of three randomly selected male canaries as a function of the total duration of songs analyzed. **B:** repertoire size in the control and ChABC-treated males before (PT) and 2 or 4 weeks after surgery (T2, T4). **C:** Number of syllables dropped at T2 or T4 as compared to the repertoire during the PT. **D:** number of syllables similar to a tutor syllable and **E:** number of added syllables that were similar to a tutor syllable at T2 and T4. $^{*}$ = $p < 0.05$ for the comparison with the control group by t-test.

syllables in the pre or post repertoire) to be lower in the ChABC group compared to the Ctrl group (Ctrl: 75.17±4.62; ChABC: 56.38±7.27; $t_{12}$ = 2.01, $p$ = 0.0676). This trend was not yet present at two weeks earlier at T2 (Ctrl: 70.00±4.22; ChABC: 72.17±1.58; $t_9$ = 0.52, $p$ = 0.6175).

**Chondroitinase ABC effectively degrades PNN in HVC.** Application of ChABC over HVC induced a dramatic decrease in the density of PNN located around PV neurons as illustrated in Fig 5.

In the 4 males that were unilaterally treated with ChABC there was a large decrease in the density of PNN in the HVC of the treated hemisphere compared to the control hemisphere one and three days after the surgery (Fig 6A). Almost no PNN remained in HVC after the treatment indicating that the enzymatic treatment actually worked. No such decrease was observed in RA and Area X (Fig 6B and 6C) confirming the localized effect of the treatment. There was also no decrease in the number of PV positive cells in the HVC nor in the RA or Area X of the treated hemisphere suggesting the treatment did not have a generalized toxic effect (Fig 6D–6F).

In subjects that were bilaterally treated with ChABC in which singing behavior was analyzed, PNN density still tended to be lower than in control birds ($t_{10}$ = 2.13, $p$ = 0.0593, Fig 6G) one month after the treatment. No changes in the density of PV interneurons could be detected ($t_{10}$ = 0.23, $p$ = 0.8246, Fig 6J). Moreover, the % PV with PNN was largely and significantly decreased by the ChABC-induced degradation ($t_{10}$ = 3.27, $p$ = 0.0085, Fig 6M). Somewhat unexpectedly, a lower PNN density ($t_{10}$ = 2.41, $p$ = 0.0364, Fig 6H) and lower % PV with PNN ($t_{10}$ = 2.30, $p$ = 0.0444, Fig 6N) associated with no change in PV neurons density ($t_{10}$ = 0.32, $p$ = 0.7589, Fig 6K) were also observed in the RA of the ChABC group compared to the controls. This difference suggests a possible transsynaptic effect of the degradation in this nucleus. No effect of the bilateral ChABC treatment was detected in area X (PNN density: $t_{10}$ = 0.14, $p$ = 0.8893; PV density: $t_{10}$ = 0.91, $p$ = 0.3839; % PV with PNN: $t_{10}$ = 0.07, $p$ = 0.9421; Fig 6I, 6L and 6O).

## Experiment 2

**Physiology and morphology.** At the beginning of the experiment in July, testosterone concentrations were low in both groups (Ctrl: 0.50±0.14, ChABC: 1.09±0.58 ng/ml; one ChABC male still had a high concentration [4.49 ng/ml], exclusion of this male decreases the group mean to 0.52±0.15) suggesting that birds had already become photorefractory, much earlier than usually observed in our laboratory. These concentrations remained low throughout the summer until males received a T-filled Silastic™ implant that induced a marked increase in circulating concentration of the steroid (Ctrl:6.99±0.79, ChABC: 5.58±0.67 ng/ml; S2A Fig). Analysis by two-way ANOVA of T concentrations in the two experimental groups at the 4 time points when measures were collected confirmed the absence of difference between groups ($F_{1,13}$ = 0.75, $p$ = 0.4026) and of groups by time interaction ($F_{3,39}$ = 1.88, $p$ = 0.1491) but there was a very pronounced effect of time ($F_{3,39}$ = 80.20, $p$<0.001). Post-hoc analysis confirmed that testosterone concentration was significantly increased at the end of the experiment compared to all previous time points ($p$<0.001 in each case; see S2 Fig for detail).

The testis mass was low at the end of the experiment and did not differ between groups ($t_{13}$ = 0.72, $p$ = 0.4833, S2B Fig) and this was also true for the syrinx weight ($t_{13}$ = 0.75, $p$ = 0.4659, S2D Fig). Somewhat surprisingly, however, the cloacal protuberance area at the end of the experiment was larger in the ChABC than in the Ctrl group ($t_{13}$ = 2.45, $p$ = 0.0290, see S2C Fig).

**Effect of ChABC on songs recorded during the next 4 weeks.** Songs recorded during the 2 pretests (PT) and the 4 weeks after surgeries were analyzed as described in experiment 1. No

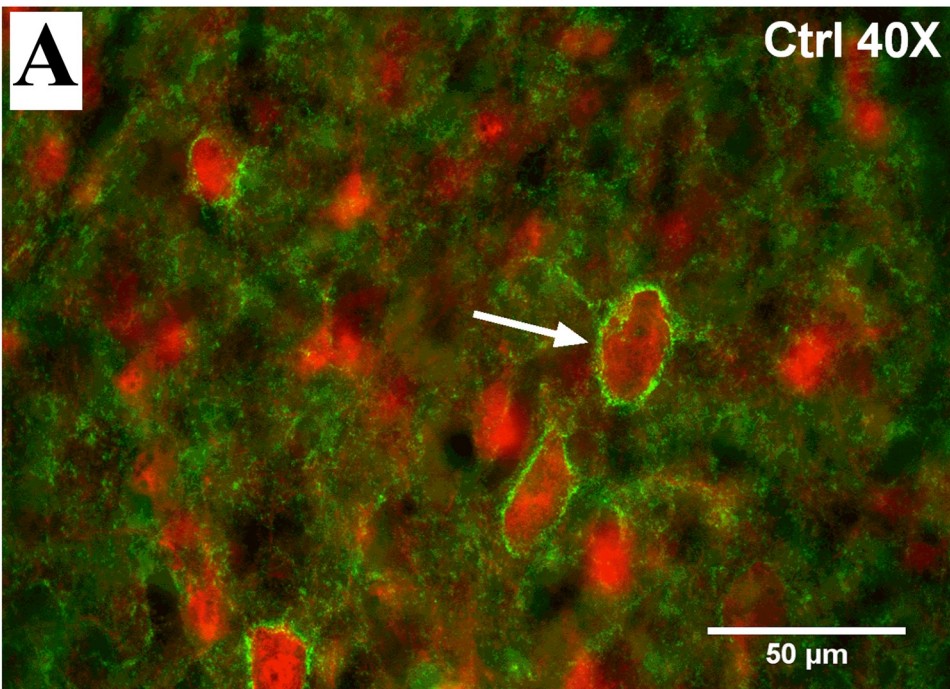

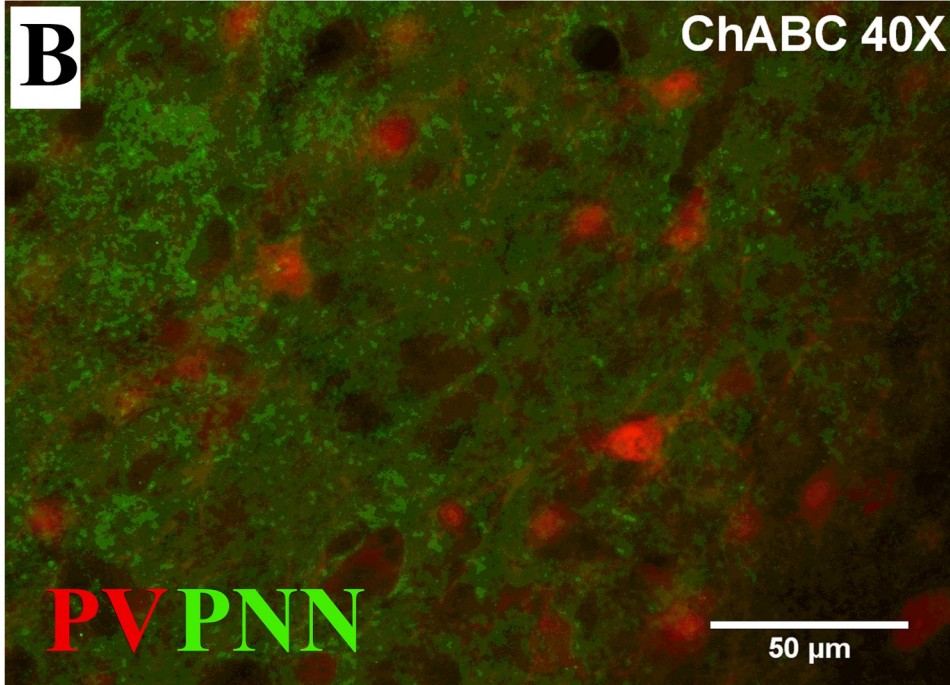

**Fig 5. Photomicrographs of PNN and PV staining in HVC.** The figure illustrates sections of a Ctrl (A) and a ChABC (B) treated male canary. PNN (green) are clearly visible around PV-positive cells (red) in the Ctrl (arrow) but not in the ChABC bird.

significant difference between groups, no effect of time and no interaction of time by group was observed in the ANOVAs of the 12 song parameters (all $p > 0.05$, see detail in S3 Fig). These analyses are however only indicative and have a very low power given the limited sample size in the Ctrl group (n = 2). Qualitative inspection of data before and after surgery in this

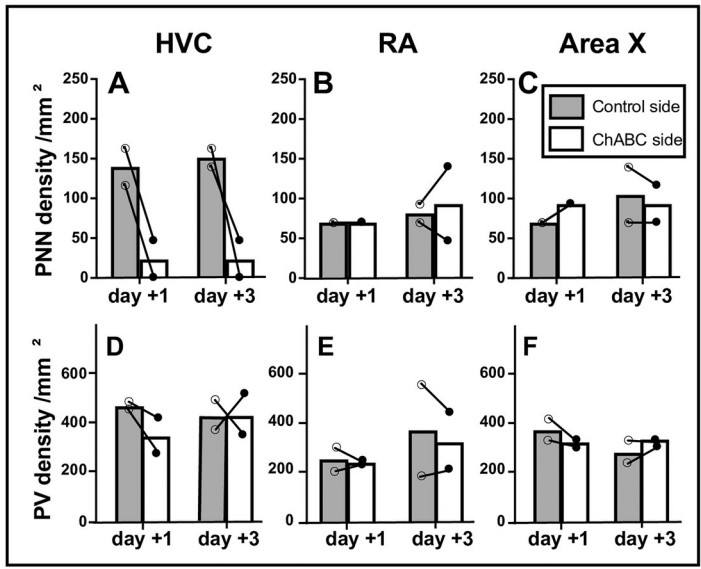

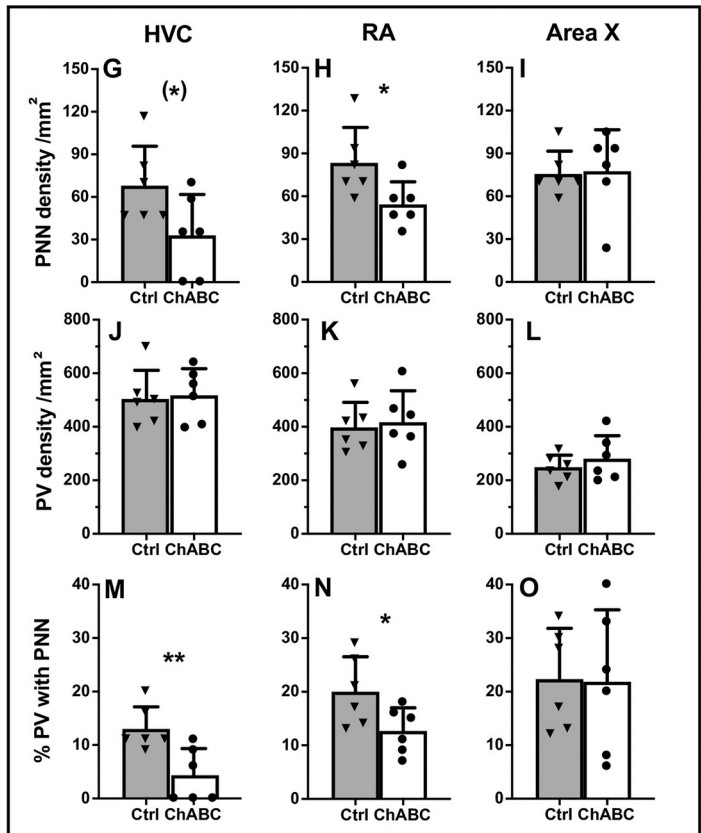

**Fig 6. Effects of chondroitinase ABC (ChABC) on PNN in the song control nuclei of male canaries. A-C:** density (numbers per mm²) of PNN and of PV in HVC (A, D), RA (B, E) and Area X (C, F) in the control (grey bars) and ChABC (open bars) hemisphere of canary brains treated unilaterally and collected 1 or 3 days after the surgery. Individual values are indicated by empty (Ctrl) or full (ChABC) dots. Connected dots represent the two hemispheres of the same bird. Bars represent the mean of each group. **G-O:** density of PNN (G-I), density of PV-positive cells (J-L), and % PV with PNN (M-O) in HVC, RA and Area X of Ctrl (grey bars) and bilaterally ChABC-treated (open bars) birds that were used to analyze singing behavior. Brains were collected one month after the treatments. Bars represent the means ± SEM and individual values are indicated by dots or triangles. T-test were used to compare groups and significant differences are indicated as follow (*) p<0.10, *p<0.05.

figure suggests however that the ChABC application had no major effect on the singing activity of the 7 birds in this group with the possible exception of some transient changes on week 1 post surgery.

**Effect of ChABC on syllables repertoire.** Repertoire size was similar in the two groups before ($t_6 = 0.50$, $p = 0.6357$) and 4 weeks after the surgery ($t_5 = 0.80$, $p = 0.4599$; Fig 7A). Comparison of the syllable repertoire before and after the surgery suggested that Ctrl and ChABC birds had dropped a similar number of syllables at T4 ($t_6 = 0.40$, $p = 0.6992$; Fig 7B). The number of syllables added between the PT and T4 was also similar in both groups ($t_6 = 0.96$, $p = 0.3742$; Fig 7C). The number of added syllables shared with the repertoire of the tutors used for playbacks was however larger in the Ctrl than in the ChABC group ($t_6 = 4.33$, $p = 0.0049$; Fig 7D). This effect must however be interpreted with caution given the small sample size.

**Singing activity induced in the fall by treatment with exogenous testosterone.** When song recordings were started again in mid September, birds were still completely silent except for two males who occasionally produced a few songs. No songs were detected in most subjects by the Matlab software in the 2 hour recordings of the 8 Ctrl and 7 ChABC males collected on 8 occasions between September 11 and October 3 suggesting that these birds were probably still photorefractory (see detailed data for September 29 and October 2, respectively Baseline (BL) 1 and 2 in Fig 8). A Silastic™ capsule filled with T was then implanted subcutaneously to all birds on October 3 and recording sessions were carried out on 13 occasions during the next 29 days before brains were collected. Ten birds were already producing songs after 2 days and singing activity progressively increased so that all subjects but one were singing by October 7. Songs recorded on 4 occasions more or less evenly spread over the next 26 days (T1 to T4, see Methods) were processed by the Matlab software and resulting data were analyzed by two-way ANOVAs (Fig 8).

After T implantation, all measures of song significantly increased over time except for song bandwidth that did not significantly vary with time ($p = 0.0740$). No overall difference between groups and no interaction between groups and time was however detected (all $p > 0.05$). After 29 days of exposure to testosterone, song rate and most song features seemed to have reached a plateau although a few characteristics such as song power were apparently still increasing.

Trills are parts of canary songs that consist short syllables quickly repeated at a constant pace. They are assumed to be one of the most difficult features of song to produce and to be particularly attractive for females. The trill rate and most of the trill characteristics changed in a drastic manner with time after birds had been treated with T (Fig 9). In particular trill rate decreased with time while their duration was increasing so that the percentage of time spent trilling during a song actually increased.

Highly significant effects of time ($p < 0.001$) were detected for each trill feature (Fig 8) except for trill entropy where a significant effect only was observed ($p < 0.05$) and for the trill mean interval duration that did not change with time ($p = 0.1903$). Again no group difference and no group by time interaction was detected (all $p > 0.05$).

**Effect on PNN degradation after 4 months.** Neuroanatomical results of this experiment are presented in Fig 9. These analyses identified no treatment effect. The density of PNN and of PV positive neurons, as well as the % PV with PNN quantified in HVC, RA or Area X were not different between Ctrl and ChABC birds (all $t_{13} < 1.36$, $p > 0.2307$ by t tests, df = 13). These results thus suggest that PNN were fully reconstructed 4 months after the surgery in this experiment (see Fig 10 for detail).

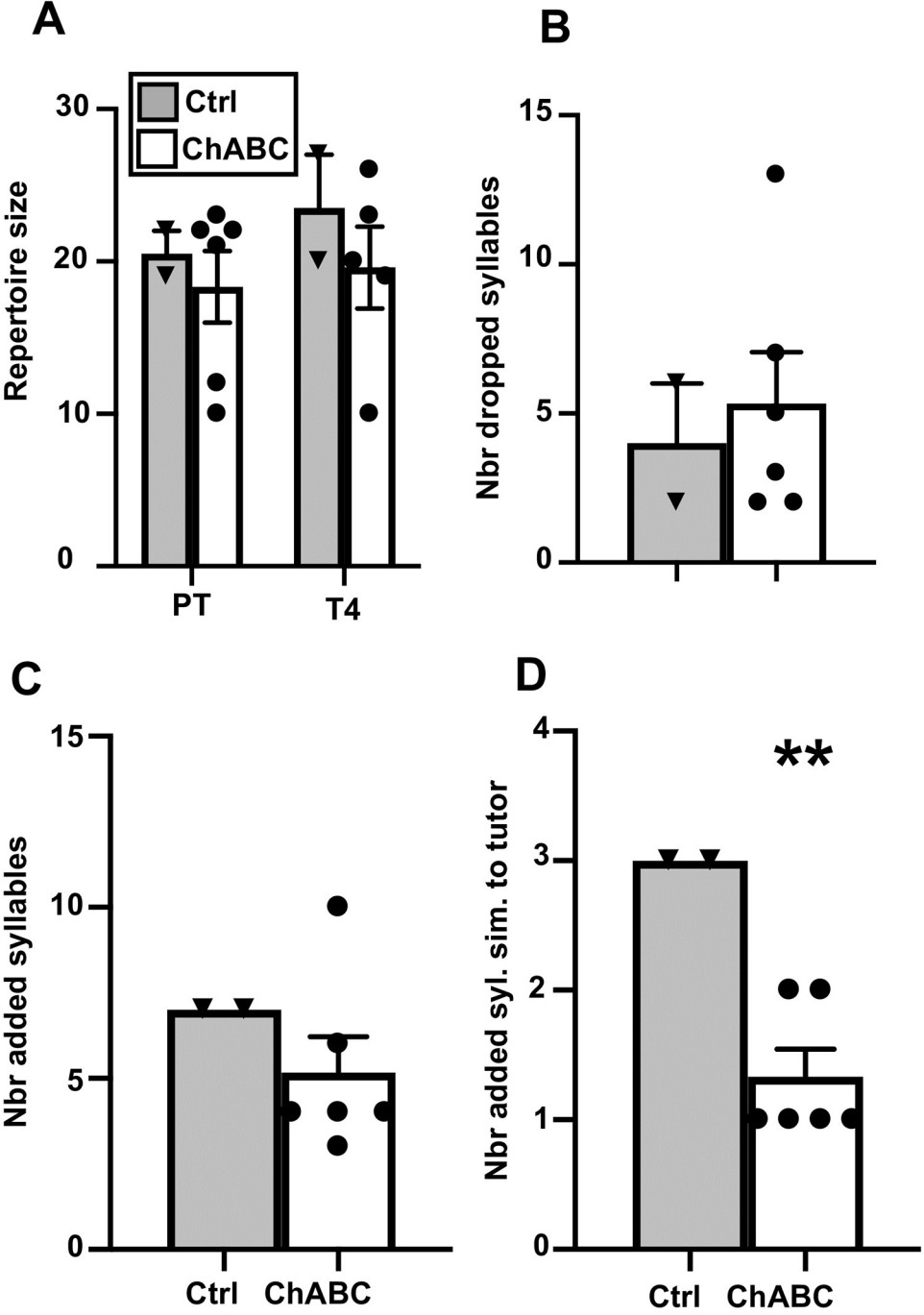

**Fig 7. Effects of ChABC on repertoire size. A.** Repertoire size in the control and ChABC-treated males before (PT) and 4 weeks after surgery (T4). **B:** Number of syllables dropped at T4 as compared to the repertoire during the PT. **C:** Number of added syllables in T4 as compared to PT. **D:** Number of added syllables in T4 as compared to PT that were similar to a tutor syllables. $^{**}$ = $p < 0.05$ for the comparison with the control group by t-test.

### Experiment 3

This 3$^{rd}$ experiment was performed at the peak of the breeding season when birds were displaying high rates of singing behavior. No treatment with exogenous T was therefore applied

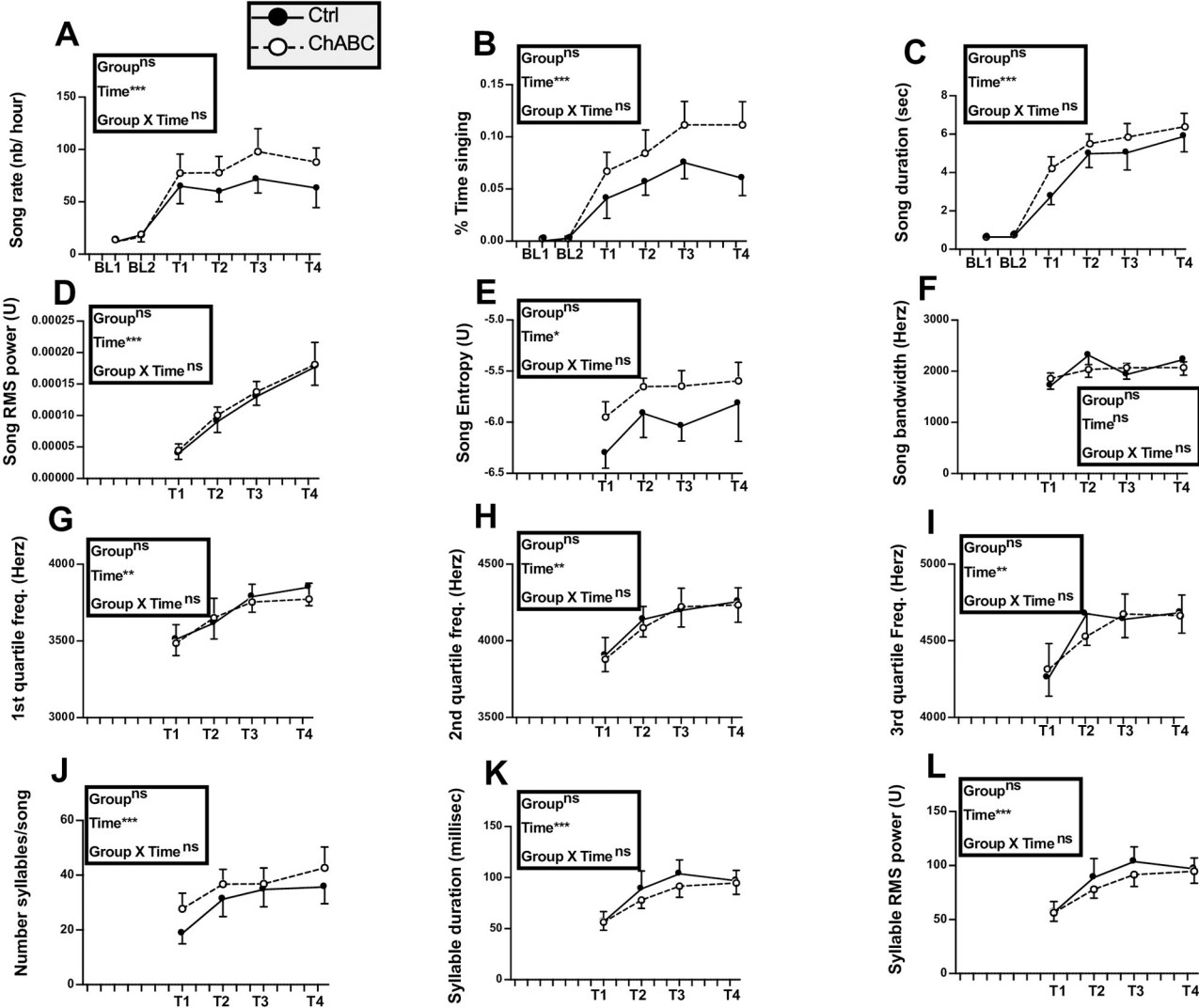

**Fig 8. Song rate and song features during the fall of experiment 2 before (Baseline, BL 1 to 2) and after (T1 to 4) all males had been treated with exogenous testosterone.** Data were analyzed by two-way ANOVA with groups as independent factor and time as a repeated factor and results are summarized in the insert for each panel. N = 8 Ctrl and 7 ChABC males.

to test whether the effects of ChABC-induced PNN degradation on singing behavior had not been obscured by the effects of T of song crystallization.

**Physiology and morphology.** Plasma T was measured in May at the beginning of the experiment and 12 weeks later in samples collected at the same time as the brains. Concentrations were as expected at the beginning of the breeding season relatively high (Ctrl: 1.20 ±0.33; ChABC: 2.01±0.83 ng/ml) and they increased slightly between May and August suggesting that the experimental procedures had no detrimental effect (Ctrl: 1.05±0.23; ChABC: 3.84±1.71 ng/ml, S4A Fig). There was no effect of time ($F_{1,20} = 0.87$, $p = 0.3611$), no interaction between time and groups ($F_{1,20} = 1.21$, $p = 0.2850$) but a tendency for group differences ($F1_{,20} = 4.17$, $p = 0.0546$). This effect was however largely driven by two males in the ChABC group who had very high T concentrations in August (5.62 and 18.09 ng/ml). After exclusion of these males the group difference was not significant (Ctrl: 1.05±0.23;

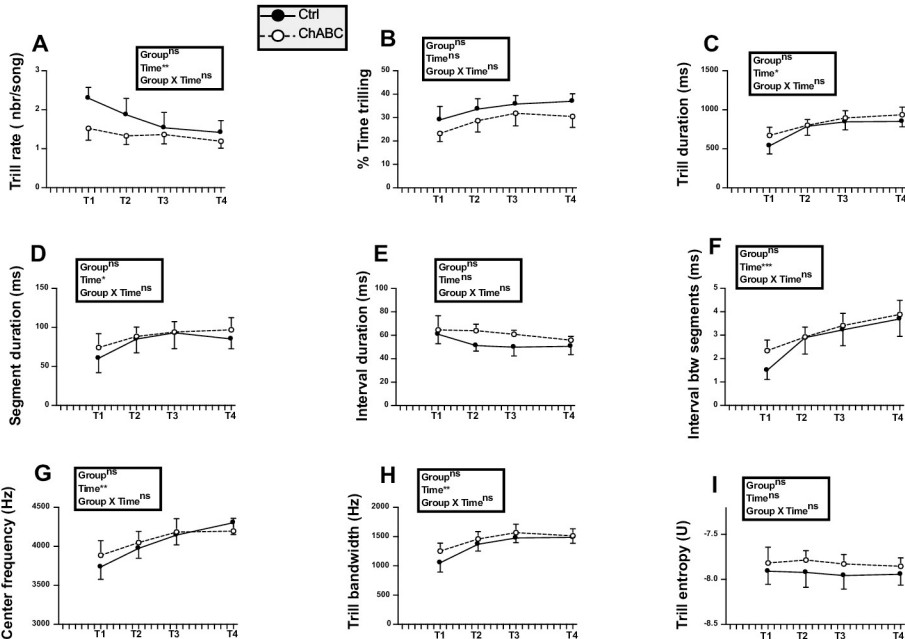

**Fig 9. Trill rate and trill features during the fall of experiment 2 after all males had been treated with exogenous testosterone (T1 to T4).** Data were analyzed by two-way ANOVA with groups as independent factor and time as a repeated factor and results are summarized in the insert for each panel. N = 8 Ctrl and 7 ChABC males.

ChABC: 1.83±.66 ng/ml; $F_{1,18}$ = 2.01, p = 0.1737) but the data from these males were retained in the analysis of other dependent variables.

No difference between groups in other measures of reproductive condition could be detected at the end of the experiment (testis mass: Ctrl:224.8±23.7; ChABC: 170.9±35.3 mg; $t_{20}$ = 1.30, p = 0.2073; cloacal protuberance area: Ctrl:38.78.±8.02; ChABC: 41.81±8.04 mm$^2$; $t_{20}$ = 0.26, p = 0.7939; syrinx mass: Ctrl:31.47±3.30; ChABC: 30.36±33.17 mg; $t_{20}$ = 0.24, p = .0.8137, S4B–S4D Fig).

**Song analysis.** Fig 11 presents the quantitative analysis of the different song features before surgeries (T0) and 2, 4, 6, 8 and 10 weeks after. As observed in the previous experiments there was a temporary modification of some song features after surgeries (e.g. decrease in % time singing, song duration, power, etc...) although this change had a smaller magnitude than in previous experiments presumably because the first recording session was performed two instead of one week later. This effect of surgery induced a significant effect of time in several of the two-way ANOVA but this effect was more moderate than in experiment 1 and males quickly recovered to sing songs that were similar to those produced before surgeries. For all song features, no overall group difference could be detected (all p>0.05). In particular, no group difference ($F_{1,13}$ = 0.16, p = 0.6942) and no group by time interaction ($F_{5,65}$ = 0.24, p = 0.9435) was affecting the average song RMS power, contrary to what had been observed in experiment 1. A single significant interaction between groups and time was present; it concerned the song entropy ($F_{5,65}$ = 2.41, p = 0.0455). Post-hoc tests, however, failed to identify any group difference at specific time points.

Analysis of trills (Fig 12) identified some changes with time but no group difference and a single group by time interaction affecting the duration of intervals between trills in a song ($F_{5,62}$ = 2.97, p = 0.0183). Post-hoc tests, however, did not detect differences between groups at any specific time point.

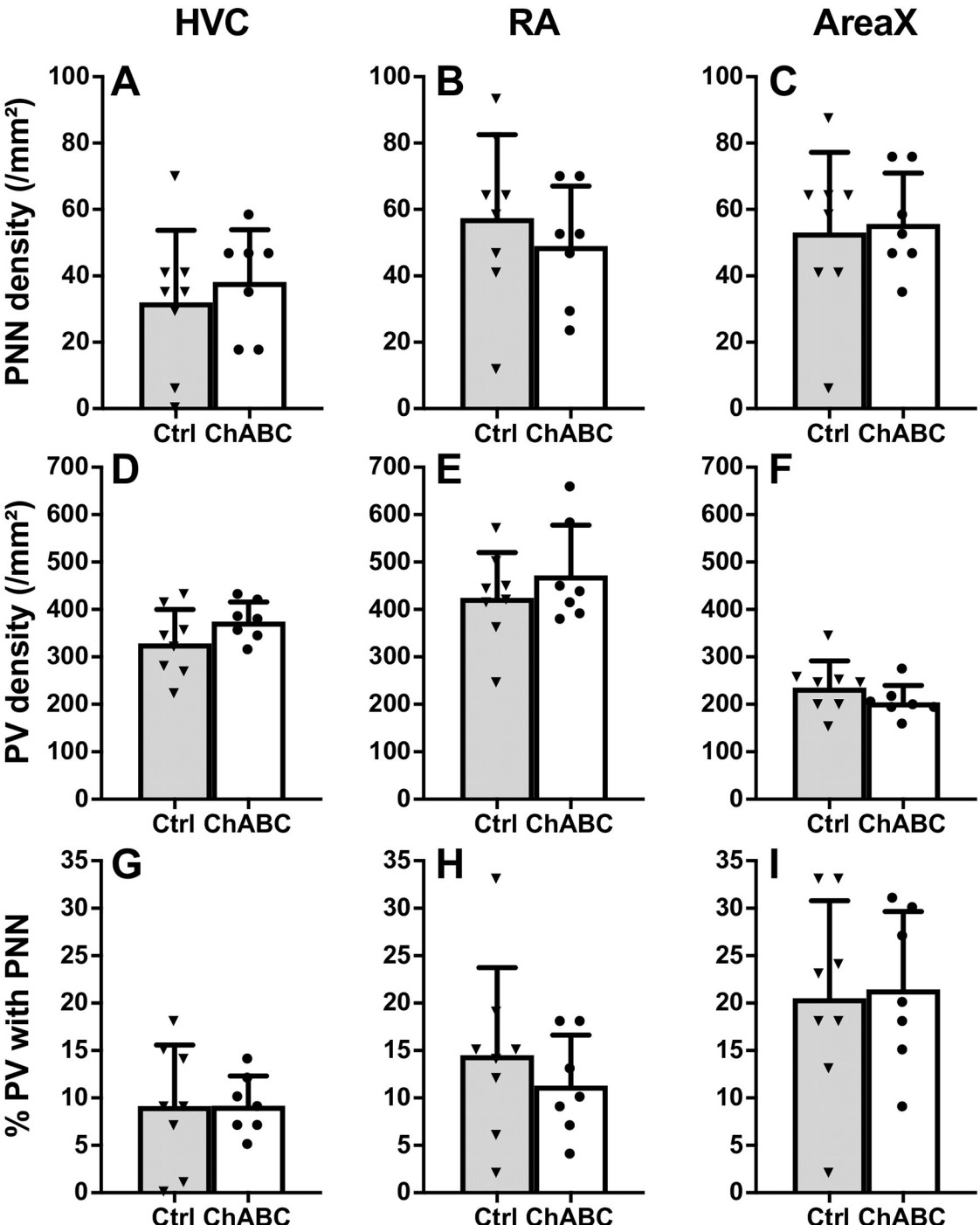

**Fig 10. Effects of chondroitinase ABC (ChABC) on PNN in the song control nuclei of male canaries as measured in experiment 2.** Density of PNN (A-C), density of PV positive cells (D-F), and % PV with PNN (G-I) in HVC, RA and Area X of Ctrl (grey bars) and bilaterally ChABC-treated (open bars) birds. Brains were collected four months after the treatments. Individual values are indicated by dots or triangles. T-test were used to compare groups but no significant differences were detected.

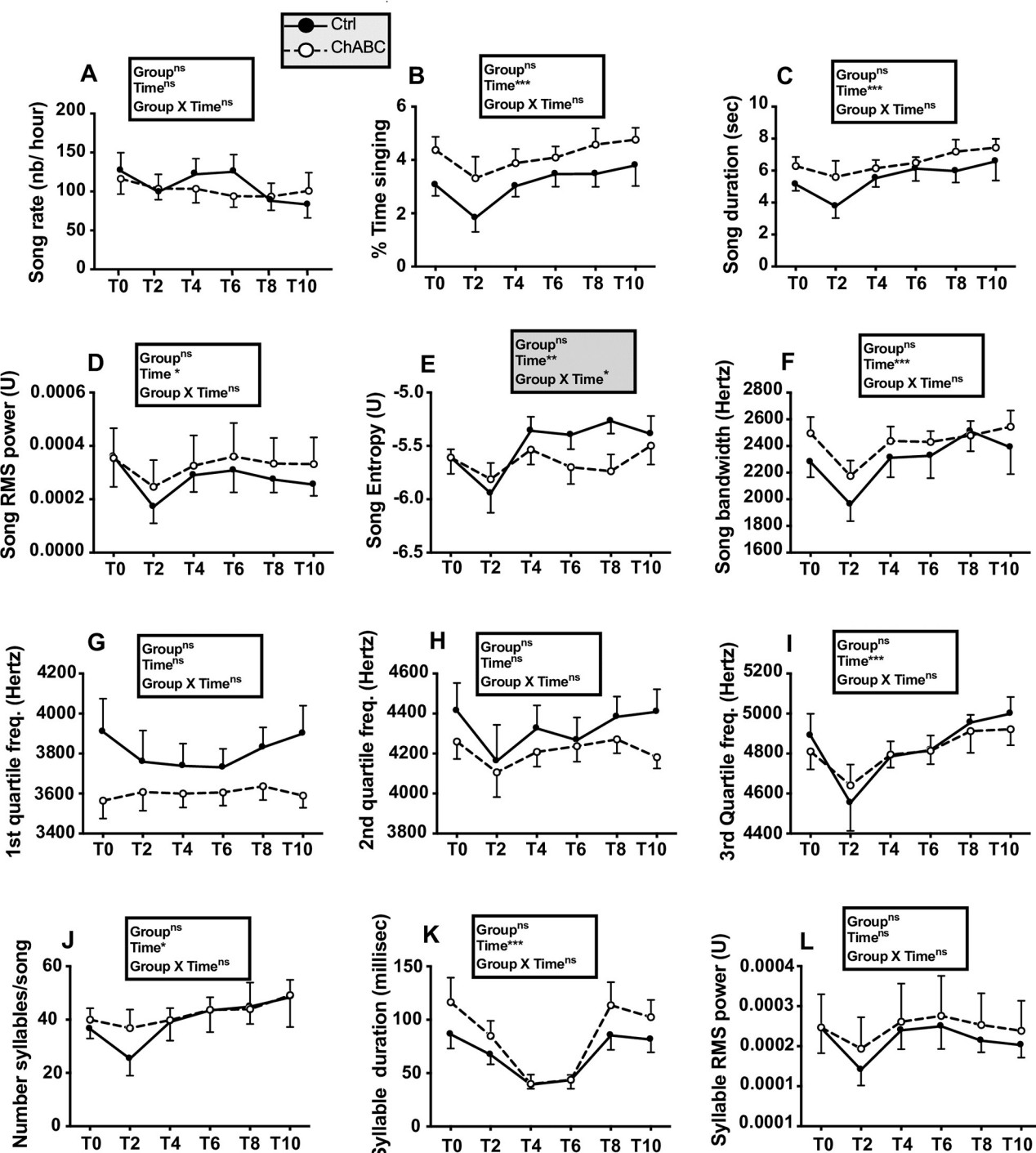

**Fig 11. Song rate and song features before (T0) and 2, 4, 6, 8 or 10 weeks after (tests T2 to T10) bilateral treatment of HVC with ChABC or the saline control.** Data were analyzed by two-way ANOVA with groups as independent factor and time as a repeated factor and results are summarized in the insert for each panel. N = 7 Ctrl and 8 ChABC males.

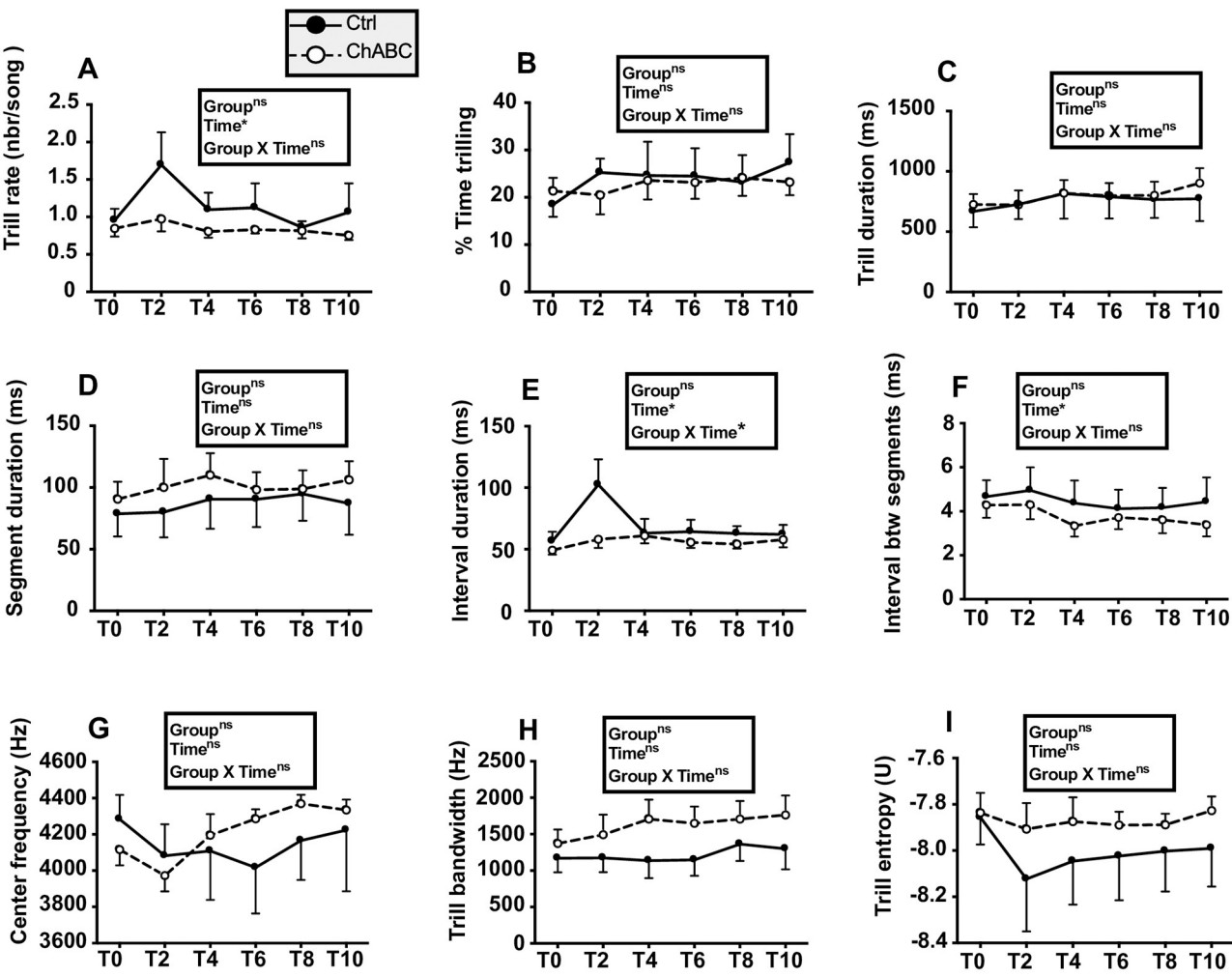

**Fig 12. Trill rate and trill features before (T0) and 2, 4, 6, 8 or 10 weeks after (tests T2 to T10) bilateral treatment of HVC with ChABC or the saline control.** Data were analyzed by two-way ANOVA with groups as independent factor and time as a repeated factor and results are summarized in the insert for each panel. N = 7 Ctrl and 8 ChABC males.

**Effects of chondroitinase ABC on PNN density.** Four months after the treatment with ChABC, there was no remaining treatment effect in HVC as observed in the previous experiment.

PNN density in ChABC males was similar to the density in Ctrl birds ($t_{13}$ = 0.41, $p$ = 0.6885, Fig 13C) and no difference in density was detected for PV interneurons ($t_{13}$ = 0.72, $p$ = 0.4852, Fig 13D) nor for the % PV with PNN ($t_{13}$ = 0.44, $p$ = 0.6636, Fig 13E). Given this lack of effects and the results of experiment 2 based on brains collected after a similar delay following surgeries, PNN in the two other song control nuclei, RA and Area X, was not analyzed.

## Discussion

This series of three experiments investigated in a variety of conditions the potential effects of the degradation of PNN in HVC on singing activity and song features of adult male canaries. These effects were assessed in the fall when the singing activity was boosted with exogenous testosterone (Experiment 1) or in the absence of exogenous testosterone when birds were still

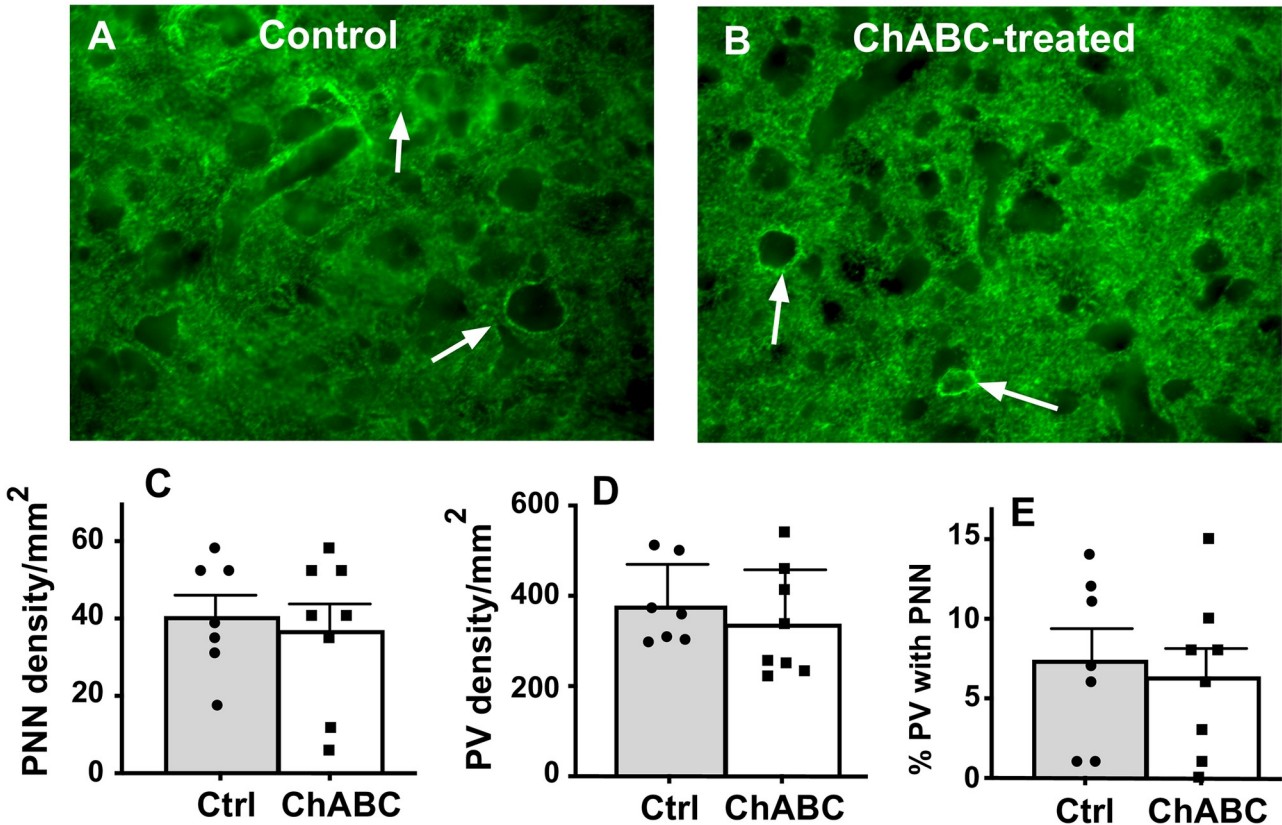

**Fig 13. The density of PNN in HVC is no longer decreased 4 months after treatment with ChABC.** Panels A and B present examples of photomicrographs illustrating the presence of PNN in the HVC of a Ctrl (A) and a ChABC-treated (B) male. Panels C to E illustrate quantification of the mean density of PNN (C), PV (D), and percentage of PV surrounded by PNN (% PV with PNN; E) in HVC, of Ctrl and ChABC-treated male canaries 4 months after the surgery. At that time no significant difference between groups could be detected.

spontaneously singing at the end (Experiment 2) and at the peak of the breeding season (Experiment 3). In an attempt to increase possible changes in song structure after PNN degradation, experimental males were exposed to the playback of unfamiliar songs types recorded from males of another breed raised in another laboratory during experiments 1 and 2. An earlier experiment on zebra finches had suggested that changes in song structure after PNN degradation are amplified when males are socially interacting with a female congener [43]. It is also known that individuals in several songbird species tend to learn songs more efficiently when exposed to live tutors as compared to taped tutoring playback experiments [48, 49]. In canaries song learning clearly occurs following tape tutoring procedures [50] but live tutoring may be even more effective. Therefore, during experiment 3 males were exposed between all recording sessions to male conspecifics producing large amounts of songs.

Despite all these variations in experimental protocols, only minimal changes in song structure were detected in these experiments. These effects require additional discussion but we would first like to consider a key methodological point concerning the efficiency of the PNN degradation procedure employed here.

## ChABC applied over HVC degrades PNN in this nucleus

ChABC application has been successfully used in a variety of mammalian studies to degrade PNN in multiple brain locations and this resulted in a variety of significant behavioral

changes within a few days or weeks (see Introduction). Two studies in zebra finches presented in abstract form only also successfully used ChABC to degrade PNN in song control nuclei [43, 44]. We confirmed here the efficacy of the procedure developed in the Teresa Nick laboratory that consists of applying on the dorsal edge of HVC a gel foam soaked with the ChABC enzyme. When applied unilaterally, the enzyme induced a unilateral marked decrease in PNN density within HVC within one day and this decrease was maintained at least three days post surgery. Some of our attempts to test with this unilateral procedure the duration of the PNN destruction produced uninterpretable results. We do not present these data here because it appears that the enzyme present in the gel foam on one side diffused to the contralateral control side that was covered with gel foam soaked in saline. The distance between the two pieces of gel foam was indeed only 3 mm and with time ChABC could diffuse through the overlying skull that is porous in birds or along the meninges.

In the main experiments testing effects on song, ChABC was applied on the dorsal edge of HVC on both sides of the brain and we could show that after one month the PNN density in HVC still tended to be lower in the treated than in the control males while the percentage of PV positive neurons surrounded by PNN was significantly smaller (Experiment 1). These effects were however no longer present when brains were collected after about 4 months in experiment 2 and 3. This finding is consistent with what has been observed in the mouse cortex where PNN that were markedly depleted by ChABC 2 weeks after injection were gradually restored over a period between 6 to 8 weeks [51].

Interestingly, there was also a decrease of PNN density and % PV with PNN in RA one month after surgery suggesting a transsynaptic effect from HVC on RA and further supporting the idea that this treatment was effective in degrading PNN in HVC (see Fig 6). It was previously shown that the seasonal growth of RA and Area X is due, at least in part, to transsynaptic effects of HVC mediated by brain-derived neurotrophic factor, BDNF [52–54]. The reduction of PNN density and of the % PV with PNN observed here in RA could be explained by a similar transsynaptic effect reflecting a change in HVC activity. However, it is also possible that the ChABC enzyme was simply transported through the $HVC_{RA}$ projection neurons.

Because HVC also sends projections to Area X, one could wonder why no changes were observed in this nucleus. This could be explained by the longer distance between these two nuclei but another explanation based on activity-dependent changes is also possible. Singing activity modulates the song control system plasticity in adult canaries [55–57]. The HVC to RA projection mediates adult singing activity whereas the projection of HVC to Area X is not directly implicated in this control [4, 58]. A change in HVC activity might thus differentially affect RA and Area X, but the nature of such a change following ChABC application remains unknown at present.

Together, these data corroborate the idea that a single application of ChABC can degrade PNN for a duration of a few weeks in birds as it does in mammals [33]. After one month however, the PNN were probably already in a process of reconstruction because the difference between treated and control birds was lower than between the treated and untreated hemisphere at 1 or 3 days after the surgery. This reconstruction process was apparently completed after 4 months and no difference could then be detected between treated and untreated birds in experiments 2 and 3. The recovery process was also possibly increased during experiment 2 by the exogenous T administration since we previously demonstrated that T increases PNN density and numbers in HVC and RA of adult canaries [42].

## Are ChABC-induced changes in song structure related to a specific physiological role of HVC in the regulation of song?

As already mentioned, the ChABC application over HVC only produced limited changes in song structure. This manipulation possibly affected song power (significant group and group by time interaction) and syllable power (significant group effect only) during experiment 1 and seemed in this experiment to be associated with a decrease in repertoire size. No significant effect was detected during experiment 2, although ChABC males added fewer syllables similar to the tutors to their repertoire than Ctrl birds. Yet this experiment had little power due to limited sample size. In the fall, exogenous testosterone also induced singing activity to the same extent in birds previously treated with ChABC and their controls. Finally experiment 3 only detected an effect of ChABC on song entropy and on the interval duration in trills (significant group by time interaction in both cases).

Given the large number of statistical tests that were carried out, these effects are possibly the result of type I statistical errors or false positive results consisting in rejecting the null hypothesis when it is in fact true (i.e., there is no difference) and they would not survive a Bonferroni or even a Benjamini-Hochberg (False Discovery Rate or FDR) correction for multiple comparisons. Yet it must also be considered that PNN degradation could mediate these effects.

In experiment 1, there was an overall treatment effect on song and syllable RMS power as well as a significant treatment by time interaction for song power. As a result, post-hoc tests identified a significant group difference for song power at all time points after, but not before, surgery. It would thus be tempting to speculate that PNN in HVC control song power.

However, a) there was already a numerical difference in song and syllables power before surgeries even if this difference did not reach statistical significance, b) this effect was not reproduced in the next two experiments, and c) most importantly, previous research never suggested that HVC might be implicated in the control of song power. It was indeed reported that the very low power of songs produced by castrated male canaries implanted with T in the preoptic area [59] is not corrected by an additional implant in HVC [60]. Furthermore, implantation of the antiandrogen flutamide in HVC did apparently not affect song power [61]. For all these reasons we are inclined to believe that these observed differences of power represent false positive effects.

By contrast, the slight modifications of song repertoire that were detected in experiment 1 (decreased repertoire size) and 2 (decreased number of added new syllables similar to tutor's syllables) might deserve further investigation. These effects had however a limited magnitude and they were observed in only a small number of subjects especially in experiment 2. In experiment 1, the decrease in repertoire size observed 4 weeks after ChABC application only concerned about 7 syllables out of 32 on average. None of the other measures of repertoire size were significantly affected. In experiment 2, ChABC males added fewer syllables similar to those of the tutor than control birds but only 2 of the controls could be studied both before and after surgeries. It should also be noted that, on a priori bases, one would rather expect that PNN degradation should increase brain plasticity and thus facilitate, not inhibit, the acquisition of new elements from the tutor song. No indication that this could be the case was detected in both experiments even if a few "new" syllables were detected in the song of both Ctrl and ChABC males.

The small number of added syllables is possibly explained by the fact that the tutor songs were from a different breed of canaries (Borders vs. Fancy Fife) and it is known that canaries preferentially imitate songs from their own over those from a different strain during developmental song learning [62]. Note also that our design did not permit us to determine when added syllables had been learned. Canaries change their song repertoire between successive

breeding seasons [14, 16] but it is not specifically known when the newly added song elements were learned. This could be during the fall and winter when song become more plastic (for data in the breed used here see: [22]) or during the early ontogeny. In white-crowned sparrows, syllables added to the repertoire during the second breeding season have indeed been heard during the earlier development [63]. To test accurately the aptitude of adult canaries to memorize new syllables following PNN degradation, it would be necessary to continuously control their acoustic environment from hatching to adulthood, which is practically difficult to achieve.

It remains that changes in repertoire size represent one of the most likely effects that would be expected following degradation of PNN in HVC. PNN limit the synaptic plasticity of PV interneurons [64, 65] that are mainly GABAergic inhibitory interneurons in HVC [66]. Inhibitory neurons are important to maintain the stability of crystallized songs [64]. In HVC they control the firing of projection neurons [67] that are selectively activated during the production of specific song elements [68] and probably control their production. Degradation of PNN in HVC should thus enhance the plasticity of interneurons and allow the rewiring of their connections with projection neurons. Assuming that different projection neurons in HVC encode for specific syllables, this process could explain modifications in the use of specific syllables.

## Limitations of the present studies: Why such limited effects

A host of studies previously demonstrated that the density of PNN in HVC correlates with the development of song and its structure across multiple conditions including sex [39, 40], age during ontogeny [37, 41], endocrine condition [42], season [22] and auditory experience [37]. This clearly suggests that PNN is in some way causally linked to some aspect of song but the present experiments largely failed to identify this link. Several reasons might possibly explain this relative failure and are considered below.

**Lack of power.** Most investigations in the present study (except part of experiment 2) were carried out in the present study with a number of subjects per group equal to 7 or 8. One could speculate that this sample size was too small to detect experimental effects. However, a power analysis (see www.gpower.hhu.de) indicates that with the repeated measures design that was implemented (song was analyzed in 15 birds on at least 4 successive occasions) our studies should detect differences with an effect size $f$ in the ANOVA equal to 0.4 (corresponding to a Cohen's d of to 0.8) with more than 90% power for a type I error (alpha) equal to 0.05. Any difference with a large effect size should thus have been detected. If the ChABC-induced PNN degradation had any effect, it should thus have a small probably biologically negligible effect size.

**Wrong features of song analyzed.** One could alternatively imagine that the song analyses that were performed here did not detect changes in song structure that could have been induced. Given the complexity of canary song, hundreds of hours of recording could indeed be analyzed only by automatic software. Based on previous work, two major types of song modifications could be expected to take place: a switch from crystallized to plastic song or the acquisition and incorporation of new song syllables. We showed previously by analyzing the Fife Fancy song during ontogeny and across seasons that plastic songs differ from crystallized songs by multiple features: they have a shorter duration, a higher entropy, a lower power and a differential distribution of energy across frequencies [22, 41].

None of these potential changes were detected. It can thus be reasonably concluded that PNN degradation in HVC is not sufficient to revert song structure from the crystallized to the plastic state. This obviously does not exclude that minute changes in the structure of some

syllables, as observed in a preliminary study of zebra finches [43] could have gone undetected. Yet such changes if they occurred must have been limited to escape detection by measures such as the entropy or energy distribution across the three quartiles of frequencies. Fine analyses at the syllables' level might possibly reveal changes that escaped detection here.

**Wrong timing for experiments.** It is established that adult male canaries sing new song elements during successive breeding seasons. Their song becomes more plastic in the late summer and fall when PNN density decreases in the song control nuclei and then in the next spring the newly crystallized song includes additional elements. Why the pharmacological degradation of PNN failed to induce a similar repertoire plasticity is unclear but could relate to the fact that PNN in HVC are only one of the factors that allow song plasticity and other factor (s) that change(s) seasonally (e.g., the photoperiod?) must be in the proper permissive condition. The three experiments were initiated at different times of the year to begin testing this hypothesis but it remains possible that none of them took place during the adequate season with the proper design. Specifically, in experiment 1 that started in November, exogenous testosterone might have inhibited plasticity normally observed in the fall since testosterone is known to promote the development of PNN in song control nuclei [42]. In experiment 2, testosterone activated in the fall the development of a similar singing activity in ChABC and Ctrl birds despite the fact that they had been exposed to the playback of unknown songs but the PNN had possibly been reconstructed at the critical time when new song was fixed. Addition of syllables could not be investigated in the 3$^{rd}$ experiment since birds had only been exposed to songs they had presumably heard before. More work should probably be invested to test this possibility.

**Wrong neuroanatomical target.** It is finally possible that degradation of PNN in HVC only is not sufficient to restore singing plasticity and a similar treatment should be applied to other song control nuclei to induce a major song reorganization. It must be noted that during the annual cycle PNN density fluctuates in the three main telencephalic song control nuclei HVC, RA and Area X and the amplitude of these changes was larger in RA and Area X than in HVC, where they were actually not statistically significant [22]. It must also be noted that Area X controls song learning and variability through inhibitory projections to DLM [69] and it might be necessary to degrade PNN in this area to induce broader effects on song structure. It might thus be necessary to degrade PNN in a bilateral manner simultaneously in HVC, RA and Area X to restore a song structure similar to the fall plastic song or to promote incorporation of new song elements.

## Conclusion

The three experiments described here identified limited effects of PNN degradation in HVC on the song structure of male canaries. These experiments establish with a fair degree of confidence that PNN in HVC are not required to maintain the general features of a crystallized song. Whether they play a role in the stability of the song repertoire cannot however be decided based on these results. Further investigations should consider producing these degradations at optimally specified time points of the seasonal cycle. It might also be necessary to develop multiple stereotaxic procedures allowing the simultaneous bilateral degradation of PNN in several song control nuclei, preferentially for longer periods. This could probably only be achieved by injecting viruses expressing the chondroitinase enzyme for extended periods as done previously in experiments performed on mammalian models [70–72]. Alternatively, given that the seasonal changes in PNN expression have a smaller amplitude than changes observed during ontogeny when the song crystallizes for the first time, one might consider testing the effects of treatments with ChABC at key steps in ontogeny.

Most importantly, one should consider that the lack of major effect in the present experiments is possibly due to the fact that no key episode of song learning took place after the degradation of PNN. Previous studies in mammals showed that degradation of PNN is able modifying the behavior resulting from a learning procedure [33, 36, 73]. Such learning could not be documented here. Future work should thus consider assessing the effects of PNN degradations in conditions where birds are experimentally induced to modify their song structure (see for example [74]).

## Supporting information

**S1 Fig. Testosterone concentrations and morphological data of Ctrl and ChABC-treated male canaries during experiment 1. A.** Testosterone concentrations before surgery (PT), after T implants (Post T), and at brain collection (End) in Ctrl and ChABC males (A). Results of the two-way ANOVA of these data are indicated in the insert (*** = p<0.001 for the comparisons with the PT time point by post-hoc tests). **B.** Testis mass, **C:** Cloacal protuberance area and **D:** Syrinx mass at brain collection in the Ctrl and ChABC groups. Individual values are presented for the last three measures that were analyzed by t-tests but indicated no significant difference.
(TIF)

**S2 Fig. Testosterone concentrations and morphological data of Ctrl and ChABC-treated male canaries during experiment 2. A.** Testosterone concentrations before surgery (PT), 5 weeks after surgery (T5), before T implants (Pre T), and at brain collection (End) in Ctrl and ChABC groups (A). Results of the two-way ANOVA of these data are indicated in the insert (*** = p<0.001 for the comparisons with all other time points by post-hoc tests). **B.** Testis mass, **C:** Cloacal protuberance area and **D:** Syrinx mass at brain collection in the Ctrl and ChABC groups. Individual values are presented for the last three measures that were analyzed by T tests (* = p<0.05).
(TIF)

**S3 Fig. Song rate and song features during two separate weeks before (pretests PT1, PT2) and the next four weeks after (tests T1 to T4) bilateral treatment of HVC with ChABC or the saline control.** Data were analyzed by two-way ANOVA with groups as independent factor and time as a repeated factor and results are summarized in the insert for each panel. These analyses identified no significant effect of the two main factors and of their interaction (all p>0.05) but have a limited power due to the reduced sample size (2 Ctrl and 7 ChABC males).
(TIF)

**S4 Fig. Testosterone concentrations and morphological data of Ctrl and ChABC-treated male canaries during experiment 3. A.** Testosterone concentrations at the beginning (Week 0) and the end (Week 12) of the experiment in Ctrl and ChABC groups. Results of the two-way ANOVA of these data are indicated in the insert (see also text for additional detail). **B.** Testis mass, **C.** Cloacal protuberance area and **D.** syrinx mass at brain collection in the Ctrl and ChABC groups. Individual values are presented for the last three measures that were analyzed by T tests but indicated no significant difference.
(TIF)

## Acknowledgments

We thank Ed Smith and Robert Dooling at the University of Maryland in College Park for designing and progressively improving the sound analysis computer program that was used to analyze recordings in this study.

## Author Contributions

**Conceptualization:** Gregory F. Ball, Charlotte A. Cornil, Jacques Balthazart.

**Data curation:** Gilles Cornez, Shelley Valle, Ednei Barros dos Santos, Ioana Chiver, Jacques Balthazart.

**Formal analysis:** Ednei Barros dos Santos.

**Funding acquisition:** Gregory F. Ball, Charlotte A. Cornil, Jacques Balthazart.

**Investigation:** Gilles Cornez, Shelley Valle.

**Methodology:** Jacques Balthazart.

**Resources:** Wendt Müller.

**Supervision:** Wendt Müller, Gregory F. Ball, Charlotte A. Cornil, Jacques Balthazart.

**Writing – original draft:** Jacques Balthazart.

**Writing – review & editing:** Gilles Cornez, Shelley Valle, Ednei Barros dos Santos, Ioana Chiver, Wendt Müller, Gregory F. Ball, Charlotte A. Cornil, Jacques Balthazart.

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
