## [Decision Letter · Decision Letter 0]

30 Jul 2021

PONE-D-21-16167

Perineuronal nets in HVC and plasticity in male canary song

PLOS ONE

Dear Dr. Balthazart,

Thank you for submitting your manuscript to PLOS ONE. After careful consideration, we feel that it has merit but does not fully meet PLOS ONE’s publication criteria as it currently stands. Therefore, we invite you to submit a revised version of the manuscript that addresses the points raised during the review process.

It is very important that you address the concerns that you provide more evidence or support that crystallized song was maintained in control birds under the same conditions, as without this evidence there could be differential interpretation of the data. 

We look forward to receiving your revised manuscript.

Kind regards,

Paul A. Bartell

Academic Editor

PLOS ONE

Journal Requirements:

Reviewers' comments:

Reviewer's Responses to Questions

**Comments to the Author**

1. Is the manuscript technically sound, and do the data support the conclusions?

Reviewer #1: Yes

Reviewer #2: Partly

Reviewer #3: Partly

2. Has the statistical analysis been performed appropriately and rigorously? 

Reviewer #1: Yes

Reviewer #2: Yes

Reviewer #3: Yes

3. Have the authors made all data underlying the findings in their manuscript fully available?

Reviewer #1: Yes

Reviewer #2: Yes

Reviewer #3: Yes

4. Is the manuscript presented in an intelligible fashion and written in standard English?

Reviewer #1: Yes

Reviewer #2: Yes

Reviewer #3: Yes

5. Review Comments to the Author

Reviewer #1: This paper reports a series of experiments that test the hypothesis that enzymatic degradation of PNNs in HVC enhances the plasticity of canaries’ songs. The authors tried several approaches to detect differences in song learning and production after PNN degradation under different seasonal conditions and both with and without testosterone treatment to promote signing behavior. The results of all of the experiments refute the hypothesis. The experiments are thoroughly described, the biochemical, histological and behavioral (song) data are carefully analyzed and appropriate statistics are used throughout. These negative results are important, because they highlight that PNN degradation, which has been reported to control specific forms of plasticity in mammals, may not be generalizable to other species and more complex forms of learning, such as sensorimotor vocal learning. The manuscript is well-written and organized.

If the immunohistochemistry images for the recovery of PNNs after 4 months are available, that would be a welcome addition to Figure 12.

A figure showing some song spectrograms would help readers understand the song analysis and demonstrate the quality of the sound recordings.

On line 719 it is mentioned that outliers in T levels were excluded from the analysis. It is not clear whether they were excluded only from the T analysis or also the measures or reproductive condition and singing behavior.

Grammar errors or typos on lines 50, 354 and 900

Reviewer #2: This paper presents a very careful and thorough description of a set of 3 experiments that generated negative (null) results. In principle, publication of null results is commendable and useful to the field. In this case, the null results imply that PNNs (in HVC) do not have a role in supporting the stability of crystallized song in canaries. This seems like it was a reasonable hypothesis based on precedents in mammals and in the development of the zebra finch song control system. All 3 experiments involved surgeries to expose HVC to an enzyme (versus control solution) to degrade PNNs. Evidence is provided that the enzyme treatment worked. For all 3 experiments, extensive quantitative analyses were conducted of the songs produced by treatment vs control groups. A possibly significant effect was only seen in one experiment on one measure (reduction of RMS power in the treated group). From this the authors derive their conclusion: "(The results) clearly establish that presence of PNN in HVC is not required to maintain general features of crystallized song"

I'm on the fence as to whether I accept this conclusion or not. The problem is that all the birds received the surgery, and the surgery alone clearly had effects, e.g., Experiment 1, line 467: "song features, however, continued to change after the surgeries in the same way in both groups" and line 487: "surgery decreased singing activity during the following weeks in all birds". And to make matters worse, in both Experiments 2 and 3, half the birds died or didn't sing. So the skeptic in me wants to ask: how do we know that singing was not already disrupted so much by the surgery (and the mites, and whatever else contributed to the death rate) that the true effect of PNN degradation was masked?

It would help some if the authors provided some comparative data from truly normal birds -- for example, sonograms showing that the lesioned birds were still producing qualitatively normal "crystallized" song compared to unlesioned birds. Without some more evidence that the control birds themselves were maintaining "general features of crystallized song", I don't see how we can conclude that additional PNN degradation had no effect.

Reviewer #3: This manuscript presents original research that I have not seen published elsewhere. The experimental design is adequate, although parts of it, especially experiment 2, are acknowledged by the authors to be underpowered. The conclusions are appropriate based on the data with appropriate caveats about not taking the conclusions (of null effects) too far. There are no concerns about ethical standard and, except for a few corrections noted below, the presentation is clear.

The goal of the experiment is laudable. They attempt to experimentally test for a causal role of perineuronal nets in canary song. These anatomical features have been shown to be correlated with features of song development, but the causal relationship is unclear. In a series of experiment, PNN were disrupted chemically and no consistent effects on song production were observed. Such null effects are important to report so that people do not overinterpret the reported correlational relationships. The one nagging difficulty is that, because of well-described problems during the experiment, some experimental power is quite low so null effects are harder to fully accept. That said, I think the authors have fairly acknowledged the limitations of the present studies and I accept the conclusion that no large effects of disrupting PNN are observed when that disruption is at the times and duration of those used in these experiments. They took a gallant stab at trying to show a causal role for PNN and finding nothing, the report will prevent others from going down the exact same rabbit-hole. They leave room and make suggestions for future experiments to test other timing, longer disruption, other areas, etc., but temporary disruption in HVC is not adequate to dramatically affect song.

Recommended changes:

Abstract: ChABC is not defined. “ …key step” should be “key steps”

Methods: “only birds that were not closely related”: Define the criterion for objectively making this judgement

Fig 2: Line breaks in 2J are in error (two breaks on one line, none on the other)

Figures involving lines: because of the low power and different group numbers, please put N in the figure captions of graphs that do not display individual data points.

L 719-720; give mean after exclusion and remove “longer” since it wasn’t initially significant either.

Fig 11: Symbols in D are in error (not open circles)

L1049 “factorsfigure 6” ?

6. PLOS authors have the option to publish the peer review history of their article (what does this mean?). If published, this will include your full peer review and any attached files.

Reviewer #1: No

Reviewer #2: **Yes: **David F. Clayton

Reviewer #3: No

---

## [Author Response · Author response to Decision Letter 0]

2 Aug 2021

Response to editorial comments and reviewers

We first would like to thank the 3 reviewers for their positive and constructive comments about our manuscript. We are now submitting a revised version that takes into account to the best of our knowledge all the critiques and suggestions that have been presented. We also took into account the editorial comments that were presented. Specifically:

-We have adapted the style of all titles and subtitles

-We include a rebuttal letter explaining how we dealt with the reviewers comments.

-All changes in the manuscript are coded in color by track changes (Revised manuscript with track-change) so that this revision can be more easily evaluated. A clean copy is provided in addition (Manuscript) that could be used for production, should this revision be accepted as it is. 

-We added a plate showing sonograms of songs in a control and ChABC-treated male to illustrate the fact that these treated males were still signing a crystallized song after the surgery. This is figure 3 in the revised manuscript and the number of each of the following figures has thus been increased by one unit.

-We modified the Funding information and Financial disclosure sections to make them match

-We added in figure 5(now figure 6) a row of bar graphs to illustrate the information (Density of PV) that was initially presented as data not shown in order to conform with the rules of the journal

Reviewer #1: This paper reports a series of experiments that test the hypothesis that enzymatic degradation of PNNs in HVC enhances the plasticity of canaries’ songs. The authors tried several approaches to detect differences in song learning and production after PNN degradation under different seasonal conditions and both with and without testosterone treatment to promote signing behavior. The results of all of the experiments refute the hypothesis. The experiments are thoroughly described, the biochemical, histological and behavioral (song) data are carefully analyzed and appropriate statistics are used throughout. These negative results are important, because they highlight that PNN degradation, which has been reported to control specific forms of plasticity in mammals, may not be generalizable to other species and more complex forms of learning, such as sensorimotor vocal learning. The manuscript is well-written and organized.

Response: Thank you very much for this summary and positive evaluation.

If the immunohistochemistry images for the recovery of PNNs after 4 months are available, that would be a welcome addition to Figure 12

Response: we have modified figure 12 to add photomicrographs of PNN 4 months after surgery in a control and a ChABC-treated male.

.

A figure showing some song spectrograms would help readers understand the song analysis and demonstrate the quality of the sound recordings.

Response: we have now added a figure containing representative sonograms of a control and a ChABC-treated male before and 4 weeks after the surgeries (Figure 3 in the revised manuscript). As can be seen the treated males still had a fully crystallized song after the treatment and they were still accurately producing the syllables they were producing during the pretests.

On line 719 it is mentioned that outliers in T levels were excluded from the analysis. It is not clear whether they were excluded only from the T analysis or also the measures or reproductive condition and singing behavior.

Response: We simply tested whether the tendency for ChABC males to have higher T concentrations remained after removal of the 2 outliers and this was not the case. These 2 outliers for T concentration were however kept for all other analyses as clearly indicated by the degrees of freedom for morphological measures discussed in the next paragraph. We added a sentence to clarify this potential uncertainly.

Grammar errors or typos on lines 50, 354 and 900

Response: these errors were corrected by adding an “s” on line 50, a comma on line 354 and deleting the word added on line 900.

Reviewer #2: This paper presents a very careful and thorough description of a set of 3 experiments that generated negative (null) results. In principle, publication of null results is commendable and useful to the field. In this case, the null results imply that PNNs (in HVC) do not have a role in supporting the stability of crystallized song in canaries. This seems like it was a reasonable hypothesis based on precedents in mammals and in the development of the zebra finch song control system. All 3 experiments involved surgeries to expose HVC to an enzyme (versus control solution) to degrade PNNs. Evidence is provided that the enzyme treatment worked. For all 3 experiments, extensive quantitative analyses were conducted of the songs produced by treatment vs control groups. A possibly significant effect was only seen in one experiment on one measure (reduction of RMS power in the treated group). From this the authors derive their conclusion: "(The results) clearly establish that presence of PNN in HVC is not required to maintain general features of crystallized song"

I'm on the fence as to whether I accept this conclusion or not. The problem is that all the birds received the surgery, and the surgery alone clearly had effects, e.g., Experiment 1, line 467: "song features, however, continued to change after the surgeries in the same way in both groups" and line 487: "surgery decreased singing activity during the following weeks in all birds". And to make matters worse, in both Experiments 2 and 3, half the birds died or didn't sing. So the skeptic in me wants to ask: how do we know that singing was not already disrupted so much by the surgery (and the mites, and whatever else contributed to the death rate) that the true effect of PNN degradation was masked?

It would help some if the authors provided some comparative data from truly normal birds -- for example, sonograms showing that the lesioned birds were still producing qualitatively normal "crystallized" song compared to unlesioned birds. Without some more evidence that the control birds themselves were maintaining "general features of crystallized song", I don't see how we can conclude that additional PNN degradation had no effect.

Response: Thank you for this accurate summary and for supporting the publication of these detailed negative results. To help reassuring this reviewer that the overall negative conclusion of our paper is really supported by data, we have added a new figure containing sonograms of songs produced by a control and a ChABC-treated male before and after the treatments. As clearly illustrated in this figure, ChABC-treated males still produced fully crystallized songs after the treatments. 

This is of course only an example but the conclusion applies for the large number of birds that successfully completed the experiment in the 3 experiments. We performed detailed quantitative analyses of the song structure in treated and control birds and no systematic difference could be detected. It must be stressed again that if song had been disrupted in any way, this would have been reflected in multiple measures such as the song entropy, the various measures of distribution of song energy, the number of syllables per song and multiple measures of fast trills.

In addition, it is certainly true that the anesthesia and surgeries transitorily disrupted song production but this only reflects an effect of stress on song production (song rate dropped markedly) but this did not affect song structure. Also quite a few birds died for various reasons but despite these problems, a very substantial number of birds completed the experiments (Expt 1: 7 Ctrl and 8 ChABC; Expt 2: 2 Ctrl and 7 ChABC; Expt 3: 7 Ctrl and 8 ChABC). This makes a total of 23 ChABC treated birds that were shown to sing fully organized songs after the treatment. So in conclusion, we are convinced that together these results strongly support the conclusions made in the manuscript that dissolution of PNN in HVC is sufficient to disorganize song in adult male canaries.

Reviewer #3: This manuscript presents original research that I have not seen published elsewhere. The experimental design is adequate, although parts of it, especially experiment 2, are acknowledged by the authors to be underpowered. The conclusions are appropriate based on the data with appropriate caveats about not taking the conclusions (of null effects) too far. There are no concerns about ethical standard and, except for a few corrections noted below, the presentation is clear.

The goal of the experiment is laudable. They attempt to experimentally test for a causal role of perineuronal nets in canary song. These anatomical features have been shown to be correlated with features of song development, but the causal relationship is unclear. In a series of experiment, PNN were disrupted chemically and no consistent effects on song production were observed. Such null effects are important to report so that people do not overinterpret the reported correlational relationships. The one nagging difficulty is that, because of well-described problems during the experiment, some experimental power is quite low so null effects are harder to fully accept. That said, I think the authors have fairly acknowledged the limitations of the present studies and I accept the conclusion that no large effects of disrupting PNN are observed when that disruption is at the times and duration of those used in these experiments. They took a gallant stab at trying to show a causal role for PNN and finding nothing, the report will prevent others from going down the exact same rabbit-hole. They leave room and make suggestions for future experiments to test other timing, longer disruption, other areas, etc., but temporary disruption in HVC is not adequate to dramatically affect song.

Response: Thank you also for these detailed and supportive comments and for supporting publication of these null results. Given the multidimensional correlation that had been established, we considered that it was important to publish these results; although some of the experiments had, as accepted in the manuscript, a low power, together we believe that they fully support our conclusion as described in more detail in the response to reviewer 2. All recommended changes have been implemented as follows.

Recommended changes:

Abstract: ChABC is not defined. “ …key step” should be “key steps”

Response: ChABC is now spelled in full and the “s” has been added to steps

Methods: “only birds that were not closely related”: Define the criterion for objectively making this judgement

Response: this sentence referred to the fact that all experimental birds came from different broods. This should have been said and this information is now added in the text.

Fig 2: Line breaks in 2J are in error (two breaks on one line, none on the other)

Figures involving lines: because of the low power and different group numbers, please put N in the figure captions of graphs that do not display individual data points.

Response: The line breaks in Figure 2J have been corrected. The numbers of subjects have also been added in the figure captions for all graphs that do not display individual data.

L 719-720; give mean after exclusion and remove “longer” since it wasn’t initially significant either.

Response: these means were added and the word longer deleted.

Fig 11: Symbols in D are in error (not open circles)

Response: these symbols have been corrected. Thank you for spotting this error.

L1049 “factorsfigure 6” ?

Response: the words “figure 6” were present by error and have been deleted.

---

## [Editor Report · Decision Letter 1]

11 Aug 2021

Perineuronal nets in HVC and plasticity in male canary song

PONE-D-21-16167R1

Dear Dr. Balthazart,

We’re pleased to inform you that your manuscript has been judged scientifically suitable for publication and will be formally accepted for publication once it meets all outstanding technical requirements.

Kind regards,

Paul A. Bartell

Academic Editor

PLOS ONE
---

## [Editor Report · Acceptance letter]

19 Aug 2021

PONE-D-21-16167R1 

Perineuronal nets in HVC and plasticity in male canary song 

Dear Dr. Balthazart:

I'm pleased to inform you that your manuscript has been deemed suitable for publication in PLOS ONE. Congratulations! Your manuscript is now with our production department. 

Kind regards, 

on behalf of

Dr. Paul A. Bartell 

Academic Editor

PLOS ONE